# GLI1 facilitates collagen-induced arthritis in mice by collaborative regulation of DNA methyltransferases

Gaoran Ge[1†], Qianping Guo[1,2†], Ying Zhou[3†], Wenming Li[1], Wei Zhang[1], Jiaxiang Bai[4], Qing Wang[1], Huaqiang Tao[1], Wei Wang[1], Zhen Wang[5], Minfeng Gan[1], Yaozeng Xu[1], Huilin Yang[1], Bin Li[1,2,6*], Dechun Geng[1*]

[1]Department of Orthopedics, The First Affiliated Hospital of Soochow University, Orthopedic Institute, Medical College, Soochow University, Suzhou, China; [2]Medical 3D Printing Center, The First Affiliated Hospital, School of Biology & Basic Medical Sciences, Suzhou Medical College, Soochow University, Suzhou, China; [3]Department of Obstetrics and Gynecology, The First Affiliated Hospital of Soochow University, Suzhou, China; [4]Department of Orthopedics, The First Affiliated Hospital of USTC, Division of Life Sciences and Medicine, University of Science and Technology of China, Anhui, China; [5]Department of Orthopaedics, Suzhou Kowloon Hospital Shanghai Jiao Tong University School of Medicine, Suzhou, China; [6]Collaborative Innovation Center of Hematology, Soochow University, Suzhou, China

*For correspondence:
binli@suda.edu.cn (BL);
szgengdc@suda.edu.cn (DG)

†These authors contributed equally to this work

Competing interest: The authors declare that no competing interests exist.

**Abstract** Rheumatoid arthritis (RA) is characterized by joint synovitis and bone destruction, the etiology of which remains to be explored. Many types of cells are involved in the progression of RA joint inflammation, among which the overactivation of M1 macrophages and osteoclasts has been thought to be an essential cause of joint inflammation and bone destruction. Glioma-associated oncogene homolog 1 (GLI1) has been revealed to be closely linked to bone metabolism. In this study, GLI1 expression in the synovial tissue of RA patients was positively correlated with RA-related scores and was highly expressed in collagen-induced arthritis (CIA) mouse articular macrophage-like cells. The decreased expression and inhibition of nuclear transfer of GLI1 downregulated macrophage M1 polarization and osteoclast activation, the effect of which was achieved by modulation of DNA methyltransferases (DNMTs) via transcriptional regulation and protein interactions. By pharmacological inhibition of GLI1, the proportion of proinflammatory macrophages and the number of osteoclasts were significantly reduced, and the joint inflammatory response and bone destruction in CIA mice were alleviated. This study clarified the mechanism of GLI1 in macrophage phenotypic changes and activation of osteoclasts, suggesting potential applications of GLI1 inhibitors in the clinical treatment of RA.

## Editor's evaluation

This study presents an important finding on the role of GLI1 in macrophages and osteoclasts in the pathogenesis of inflammatory arthritis, and suggests a therapeutic potential of GLI1 targeting in rheumatoid arthritis. The evidence supporting the claims of the authors is solid, although inclusion of the analysis of cell type-specific GLI1-deficient mice would have strengthened the study. The work will be of broad interest to immunologists, rheumatologists and bone biologists.

## Introduction

Rheumatoid arthritis (RA) is a common chronic inflammatory disease that currently affects approximately 75 million people worldwide. In addition to arthritis symptoms, RA can eventually lead to disability or even death (*Firestein, 2003*; *Kim et al., 2019*). Although the mechanism of RA is not clear, it is generally believed that synovial inflammation and bone erosion are the direct factors causing joint damage. Therefore, unraveling the molecular pathways that underlie immune regulation and bone destruction is of major interest to better understand the pathophysiology of RA and to design new approaches to achieve a therapy for this severe joint disease.

Persistent activation of immune cells leads to the progression of symptoms such as synovitis in RA. This immune-mediated disorder involves both innate and adaptive cellular compartments as well as their dysregulated cytokine production. In addition, the pathological process of RA is promoted through the synergistic action of the cellular resident in the bone and in joint compartments, such as osteoclasts, chondrocytes and stromal cells (*Komatsu and Takayanagi, 2022*). Within the innate immune function that causes inflammation, macrophages have a key role. In fact, they contribute to normal tissue homeostasis, working as antigen-presenting cells (APCs) to activate adaptive immunity, pathogen expulsion, resolution of inflammation and wound healing (*Tardito et al., 2019*). Classically activated macrophages (M1), which have been proven to be dominant in RA joints, cause joint erosion, secreting principally proinflammatory cytokines such as tumor necrosis factor α (TNF-α), interleukin 1β (IL-1β) and IL-6, whereas alternatively activated macrophages (M2) contribute to tissue remodeling and repair via high production of anti-inflammatory cytokines (mainly IL-10 and TGF-β) (*Cho et al., 2014*; *Tardito et al., 2019*; *Wu et al., 2020*). Another hallmark is irreversible bone destruction, which is led by osteoclasts. The RA microenvironment and proinflammatory cytokines stimulate the formation of osteoclasts, which have long been considered the only cells capable of absorbing bone matrix, and the subsequent degradation of bone and cartilage (*Bustamante et al., 2017*; *Woo et al., 2018*; *Yang et al., 2019*). Consequently, effectively hindering the overactivation of osteoclasts is one of the keys to alleviating excessive bone resorption.

The hedgehog pathway, which has been shown to be closely involved in osteogenesis (*Jemtland et al., 2003*), is highly conserved and critical for normal embryogenesis. The hedgehog protein family consists of hedgehog ligands that bind the cell surface transmembrane receptor patch (PTCH) and function through posttranslational processing of glioma-associated oncogene homolog (GLI) zinc finger transcription factors (*Rimkus et al., 2016*; *Sasai et al., 2019*). To date, three mammalian GLI proteins have been identified, among which GLI1 usually acts as a transcriptional activator. On the basis of these studies, small molecular compounds such as GANT58 (selective inhibitor of GLI1) and GANT61 (inhibitor of GLI1 and GLI2) are often used as pharmacological interventions of GLI1, so as to achieve the purpose of inhibiting GLI1 activity and regulating the molecular biological processes (*Chen et al., 2018*; *Schneider et al., 2018*). Many of the physiopathological processes involved with GLIs are complex and worth discussing. Relevant studies have shown that GLI1-activated transcription promotes the development of inflammatory diseases such as gastritis, and antagonizing GLI1 transcription can alleviate the inflammatory degradation of articular cartilage (*Ali et al., 2016*; *Kopinke et al., 2021*). Accordingly, these clues suggest that GLI1 may be involved in inflammation and bone erosion processes, driving us to explore its role and regulatory mechanisms in RA.

DNA methylation, which is activated by DNA methyltransferases (DNMTs), is an important epigenetic marker that plays an important role in regulating gene expression, maintaining chromatin structure, gene imprinting, X chromosome inactivation and embryo development (*Li and Zhang, 2014*). As reported, DNMT1 and DNMT3a are involved in the progression of many physiological disorders, such as the immune response and cell differentiation (*Fu et al., 2022*; *Ramabadran et al., 2023*). In this study, we investigated the association of GLI1 with RA as well as the critical role of GLI1 in the pathological process of joint inflammation and bone destruction. It was confirmed that inhibition of GLI1 could reduce inflammatory bone destruction in collagen-induced arthritis (CIA) mice. Mechanistically, the intranuclear translocation of GLI1 was shown to promote M1 polarization of macrophages and overactivation of osteoclasts by collaborative regulation of DNMTs. In general, this study not only clarified the pivotal role and mechanism of GLI1 in joint inflammation and bone destruction in RA but also suggested potential applications of GLI1 inhibitors in clinical treatment.

## Results

### GLI1 expression is elevated in human RA synovium, and a selective inhibitor of GLI1 can alleviate joint bone destruction in CIA mice

To determine the relationship between GLI1 expression and the RA process, we collected synovial tissue from RA patients and noninflammatory joint disease donors and obtained informed consent from the patients. Compared with normal synovial tissue, the synovium of RA patients had a high degree of inflammatory cell infiltration (*Figure 1A*). Immunohistochemical (IHC) staining showed that there were more GLI1-positive cells in RA synovium than in normal synovium (*Figure 1B and C*). The results of western blotting further verified the upregulation of GLI1 in the joint synovium of RA patients (*Figure 1—figure supplement 1A–C*). We performed a DAS28 score on the included patients to initially assess RA activity and then matched the relative expression of GLI1 with them, indicating that GLI1 was more highly expressed in RA patients. These results were of concern to us and implied that GLI1 plays a catalytic role in the pathological process of RA.

To verify this speculation, we constructed a collagen-induced arthritis (CIA) model using DBA mice and performed a therapeutic intervention with the GLI1 selective inhibitor GANT58. All mice survived and were included in the analysis. H&E staining results showed no hepatorenal toxicity of GANT58 (*Figure 1—figure supplement 2*). After secondary immunization, CIA mouse paws swelled rapidly, and weight loss occurred simultaneously, which could be reversed by GANT58 treatment (*Figure 1—figure supplement 3A, B*). Based on the arthritis score, it could be observed that after 21 days of modeling, the scores of the mice in the vehicle group increased significantly. Obviously, the arthritis scores of GANT58-treated mice were remarkably lower than those of mice in the vehicle group (*Figure 1D*). We photographed the hind paws of mice in each group and found that CIA mice had obvious signs of arthritis, while mice in the GANT58 treatment group showed a slight degree of swelling, which was similar to that of normal mice (*Figure 1E*). Micro-CT was performed to scan the mouse paws and ankle joints. As shown in *Figure 1F*, compared with sham group mice, vehicle group mice had more obvious bone destruction, and GANT58 treatment alleviated this effect. Through the analysis of bone parameters in the joint part of the mouse foot claw, we also found that the bone parameters were markedly better in the GANT58-treated mice than in the vehicle control mice (*Figure 1G*). Based on these results, we performed histological sectioning and staining of mouse paw articular tissue as well as knee tissue to evaluate pathological histological changes in the joints. The results showed that GANT58 intervention significantly reduced the degree of inflammatory cell infiltration and joint bone destruction (*Figure 1H, I, Figure 1—figure supplement 3C, D*). This evidence suggested that inhibiting the activity of GLI1 significantly reduced the occurrence and development of arthritis in CIA mice. However, it is not clear what the specific regulatory effect of GLI1 is. Thus, we performed immunofluorescence colocalization staining of the macrophage marker F4/80 and GLI1 both in subchondral bone marrow cavity and synovial tissue and observed that the high expression of GLI1 in CIA mice was mostly located in macrophage-like cells (*Figure 1J*). Therefore, our follow-up research mainly focused on the mechanism of GLI1 in the regulation of macrophage fate.

### GLI1 affects the release of inflammatory cytokines by regulating the macrophage phenotype

To investigate the relationship between GLI1 expression and M1 macrophages in vitro, we first measured GLI1 protein expression by western blotting during M1 macrophage induction, which increased at 24 hr after LPS and IFN-γ induction (*Figure 2A and B*). Typically, when the GLI1-related pathway is activated, GLI1 is transported from the cytoplasm to the nucleus, which in turn exerts its transcriptional regulatory role. Therefore, we separately examined the amount of GLI1 protein inside and outside the nucleus. As shown in *Figure 2A*, after LPS/IFN-γ intervention, although the amount within the cytoplasm did not change significantly, the expression of GLI1 in the nucleus increased to certain degrees, suggesting that GLI1 was activated and incorporated into the nucleus during the M1 polarization process to play a related regulatory role.

Given this finding, we further utilized GANT58 to intervene in the macrophage polarization process. Cell viability was measured by CCK-8 assay before the in vitro use of GANT58, and the results showed that GANT58 had no significant inhibitory effect on either BMMs or RAW264.7 cells until the concentration reached 40 μM (*Figure 2—figure supplement 1A–D*). We treated cells with a concentration of

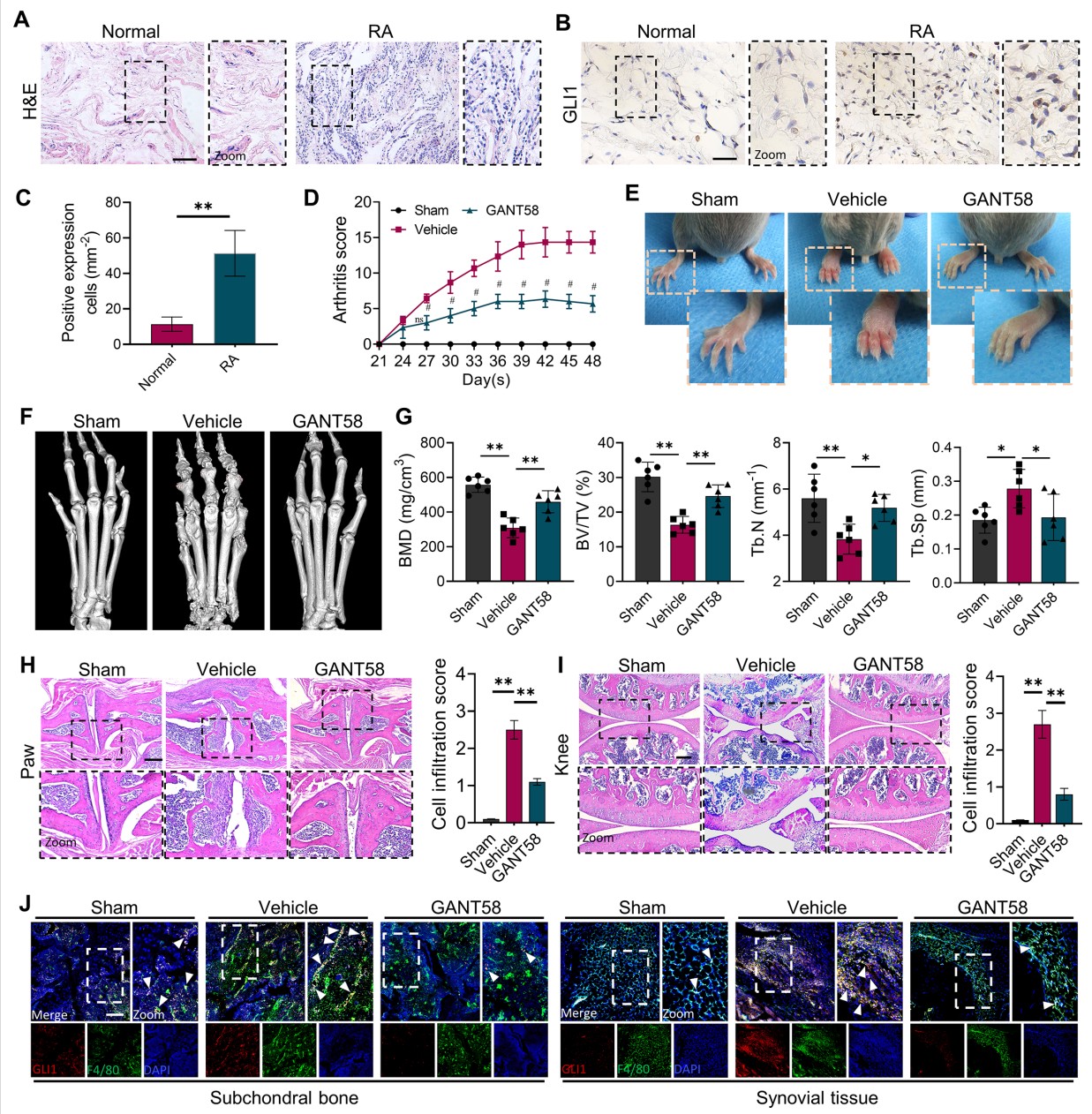

**Figure 1.** GLI1 is highly expressed in human RA synovium and regulates the pathological process of joint inflammatory bone destruction in CIA mice. (**A**) H&E staining of normal and RA human synovium. Scale bars = 200 μm. (**B**) IHC staining of GLI1 in normal and RA human synovium and (**C**) quantification of positively stained cells. Scale bars = 50 μm. n=3. (**D**) Arthritis score of mouse limbs. (**E**) Photos of mouse paws. (**F**) Micro-CT scanning and 3D reconstruction of mouse paws. (**G**) Bone parameters of BMD, BV/TV, Tb. N, Tb.Th. (**H**) Images of H&E staining of mouse paw joints and inflammatory cell infiltration score knee joints. Scale bars = 200 μm. (**I**) Images of H&E staining of mouse knee joints and inflammatory cell infiltration score. Scale bars = 200 μm. (**J**) Immunofluorescence costaining image of F4/80 with GLI1 (red: GLI1, green: F4/80, blue: DAPI). Scale bars = 200 μm. The in vivo results are presented as the mean ± SD of 6 mice per group. Data shown represent the mean ± SD. Statistical analysis was performed using one-way ANOVA test. *p<0.05, **p<0.01.

The online version of this article includes the following source data and figure supplement(s) for figure 1:

**Figure supplement 1.** GLI1 is highly expressed in synovial tissue of rheumatoid arthritis patients.

**Figure supplement 1—source data 1.** Uncropped western blot images for *Figure 1—figure supplement 1*.

**Figure supplement 2.** GANT58 shows no obvious toxicity in CIA mice.

**Figure supplement 3.** GANT58 reduces arthritis symptoms and bone destruction in CIA mice.

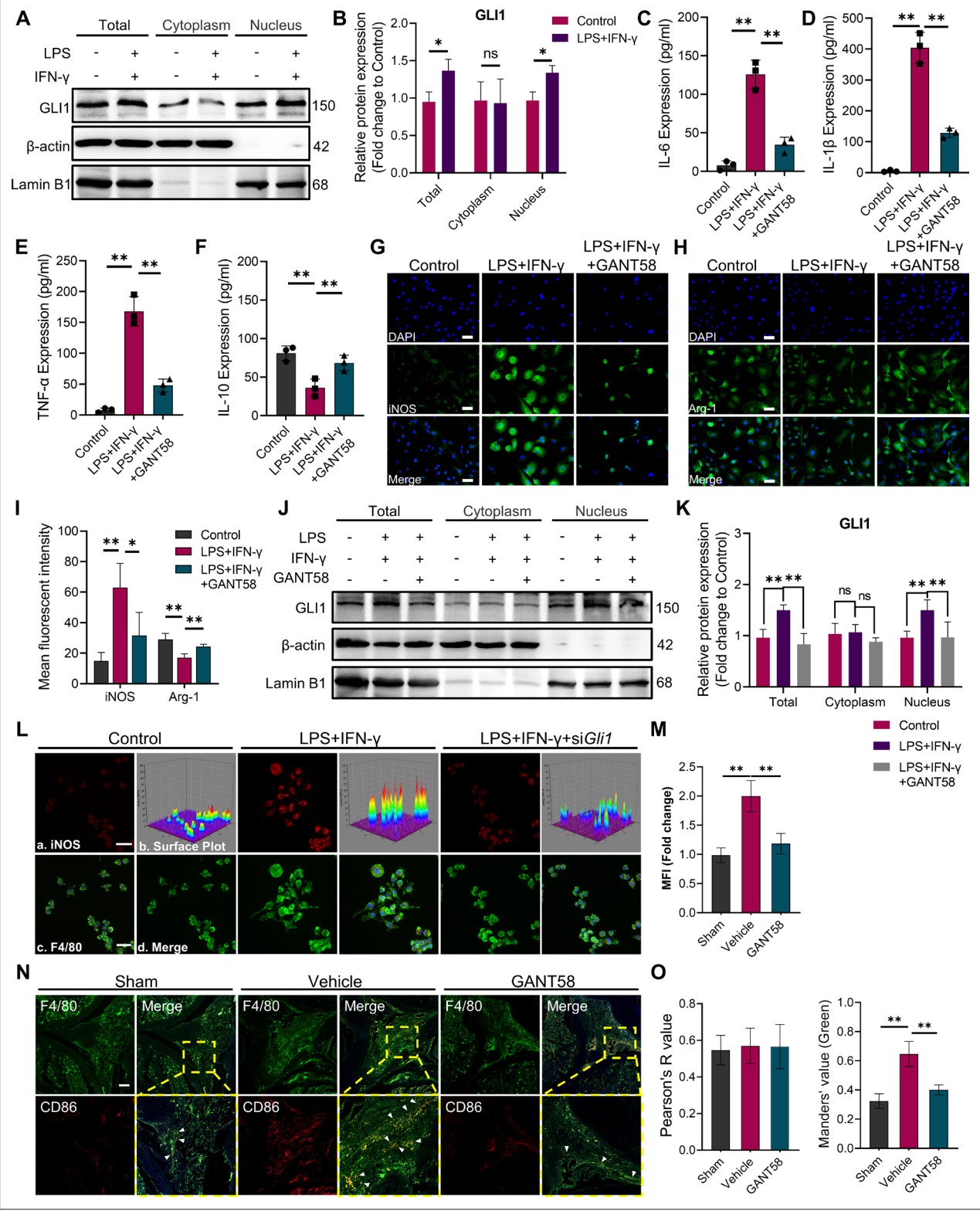

**Figure 2.** GLI1 plays an active role in M1 macrophage formation and the release of proinflammatory cytokines. (**A**, **B**) RAW264.7 cells were stimulated with LPS/IFN-γ for 24 hr. The total proteins and proteins in the cytoplasm and nucleus were isolated and extracted. GLI1 protein was detected by western blot, and the grayscale value ratio to β-actin of the western blot results was analyzed. n=3. Statistical analysis was performed using Student's t test. (**C–F**) Macrophages were stimulated with LPS/IFN-γ for 36 hr, with or without 10 μM GANT58 treatment. The supernatant was collected, and the volumes of IL-1β, IL-6, TNF-α and IL-10 were detected by ELISA. (**G, H**) Immunofluorescence staining of iNOS and arginase (Arg-1) after 24 hr of LPS/

*Figure 2 continued on next page*

*Figure 2 continued*

IFN-γ induction, with or without 10 µM GANT58 treatment. Scale bars = 25 µm. (**I**) The mean fluorescence intensity of immunofluorescence was analyzed using ImageJ. (**J**, **K**) RAW264.7 cells were stimulated with LPS/IFN-γ for 24 hr with or without 10 µM GANT58 pretreatment. GLI1 protein was detected by western blot, and the grayscale value ratio to β-actin of the western blot results was analyzed. n=3. (**L**) Immunofluorescence staining of iNOS in WT and si*Gli1* RAW264.7 cells after LPS/IFN-γ induction for 24 hr. Scale bars = 20 µm. (**M**) The mean fluorescence intensity of immunofluorescence was analyzed using ImageJ. (**N**) Immunofluorescence staining of mouse joint tissue (green: F4/80, red: CD86). Scale bars = 200 µm. (**O**) Pearson colocation coefficient and CD86-positive quantitative analysis were performed with ImageJ. Statistical analysis was performed using one-way ANOVA test. Data shown represent the mean ± SD. *p<0.05, **p<0.01, ns = no significance.

The online version of this article includes the following source data and figure supplement(s) for figure 2:

**Source data 1.** Uncropped western blot images for *Figure 2*.

**Figure supplement 1.** Cytotoxicity detection of GANT58.

**Figure supplement 2.** GLI1 protein expression after GANT58 and siRNA treatment.

**Figure supplement 2—source data 1.** Uncropped western blot images for *Figure 2—figure supplement 2*.

**Figure supplement 3.** GANT58 inhibits M1 macrophage polarization.

**Figure supplement 3—source data 1.** Uncropped western blot images for *Figure 2—figure supplement 3*.

10 µM and found that it reduced the expression of GLI1 (*Figure 2—figure supplement 2A, B*). Subsequently, we performed GANT58 intervention on M1-induced macrophages, collecting and detecting changes in inflammatory cytokine content in the cell supernatant. Stimulated by LPS and IFN-γ, macrophages release a large number of proinflammatory cytokines, including IL-1β, IL-6, and TNF-α. After the inhibitory intervention of GLI1, the content of these cytokines was significantly reduced, while the release inhibition of IL-10 was alleviated to some extent (*Figure 2C–F*). Immunofluorescence staining suggested that the GANT58 group exhibited fewer iNOS-positive cells and near-basic levels of Arg-1 expression compared to cells in the LPS and IFN-γ-treated groups (*Figure 2G–I*), and the western blot results supported the same conclusion (*Figure 2—figure supplement 3A, B*). To rule out the change in cytokine release caused by the change in cell number, we also detected the expression level of proinflammatory cytokine mRNA. The results showed that GANT58 also reduced the expression of the *Il1b*, *Il6* and *Tnfa* genes (*Figure 2—figure supplement 3C*).In addition, we examined the expression and distribution of GLI1 in cells under different interventions and found that GANT58 significantly reduced the nucleation transport of GLI1 while inhibiting the proinflammatory state of cells (*Figure 2J and K*). According to the above results, we knocked down *Gli1* by siRNA to further verify the specificity of GLI1 in regulating the polarization of macrophages (*Figure 2—figure supplement 2A, B*, *Supplementary file 1*). According to the results of western bolt, the si*Gli1* #1 was used for *Gli1* knockdown in the next experiments. As shown in *Figure 2L and M*, compared with wild-type (WT) macrophages, si*Gli1* macrophages showed lower iNOS expression induced by LPS and IFN-γ. Subsequently, we performed immunofluorescence co-staining of F4/80 and CD86 in animal model tissues, and compared with more CD86-positive M1 macrophages in the synovial tissue of mice in the CIA model group, the number of positive macrophages of the proinflammatory phenotype after GANT58 treatment was significantly reduced (*Figure 2N and O*). These results confirmed that GANT58 was capable of reducing the maturation of M1 macrophages and the release of proinflammatory cytokines by affecting the activation of GLI1 both in vitro and in vivo.

## The expression and intranuclear transport of GLI1 is involved in osteoclast activation

The overactivation of osteoclasts is the direct cause of bone destruction in RA. As described in the in vivo experimental results in the first part, we found that GLI1 is highly expressed in macrophage-like cells in the subchondral bone of the joints, which raised our concerns about GLI1 and osteoclasts. To investigate the role of GLI1 in osteoclast formation, we measured GLI1 protein expression in RAW264.7 2 days after RANKL induction. The results showed that after RANKL stimulation, more GLI1 was activated and transferred to the nucleus (*Figure 3A and B*). Thus, to confirm the regulatory function of GLI1 in the differentiation of osteoclasts, we used GANT58 to intervene in GLI1 activity during RANKL induction. Primary BMMs were isolated and cultured with M-CSF, followed by stimulation with RANKL, with or without 6hr of GANT58 pretreatment. As a result, there was less osteoclast formation in the GANT58-treated group than in the RANKL stimulation group (*Figure 3C and D*). To eliminate

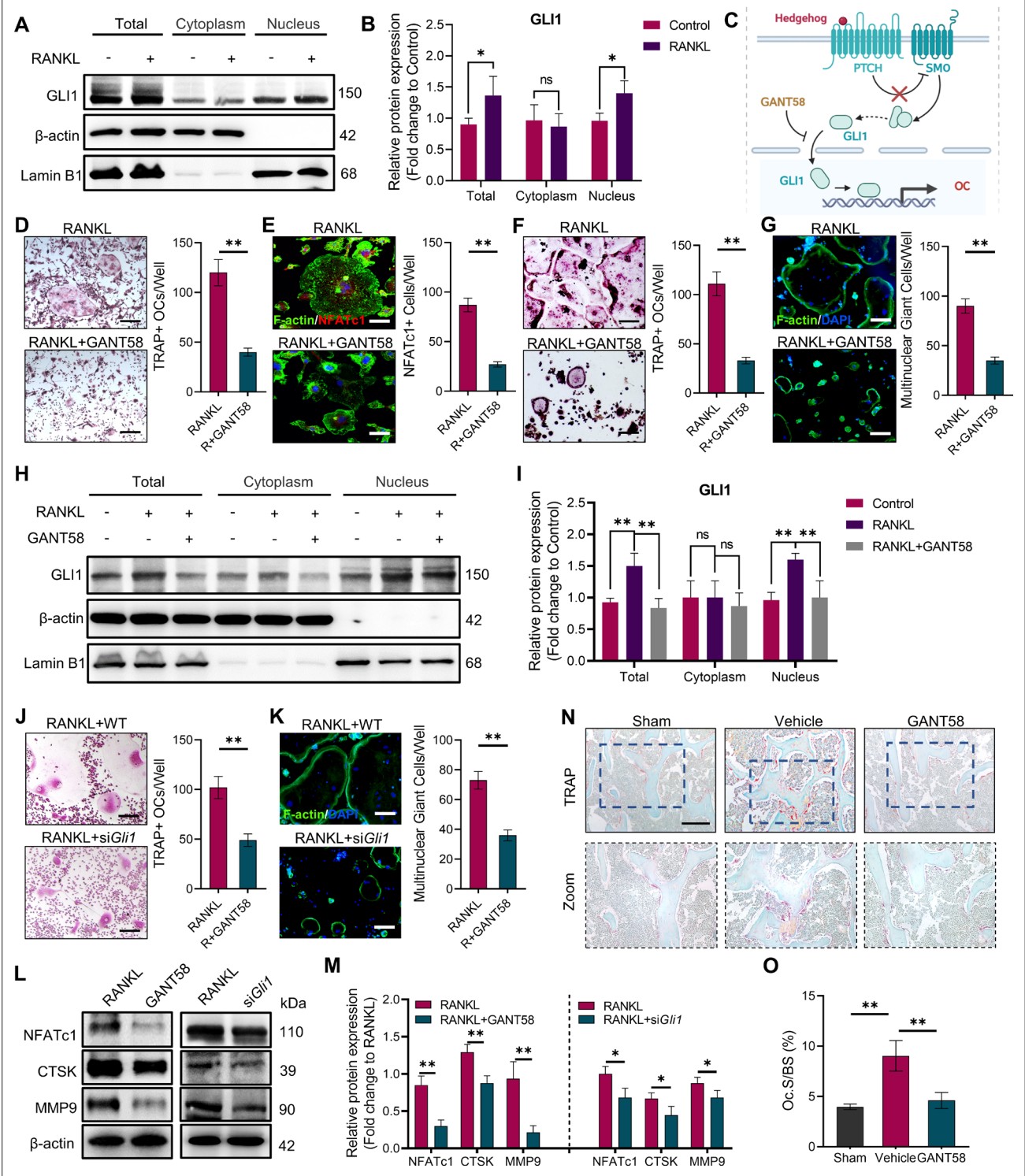

**Figure 3.** GLI1 regulates the formation of osteoclasts in vivo and in vitro. (**A, B**) RAW264.7 cells were stimulated by RANKL for 48 hr. Total GLI1 protein and GLI1 in the cytoplasm and nucleus were detected by western blotting during osteoclast induction, and the fold change in the grayscale value compared to the control group was analyzed. n=3. (**C**) Schematic diagram of GLI1 activation. (**D**) TRAP staining of BMMs with RANKL induction and TRAP-positive osteoclast number per well. Scale bars = 25 μm. (**E**) F-actin and NFATc1 immunofluorescence staining of RANKL-stimulated BMM-derived osteoclasts and NFATc1-positive cell quantity per well. Scale bars = 25 μm. (**F**) TRAP staining of RAW264.7 cells with RANKL induction and TRAP-positive osteoclast quantity per well. Scale bars = 25 μm. (**G**) F-actin immunofluorescence staining of RANKL-stimulated RAW264.7-derived osteoclasts and multinucleated giant cell quantity per well. Scale bars = 25 μm. (**H, I**) RAW264.7 cells were stimulated by RANKL for 48 hr. Total GLI1 protein and GLI1 in

*Figure 3 continued on next page*

*Figure 3 continued*

the cytoplasm and nucleus were detected by western blot during osteoclast induction treated with or without GANT58, and the fold change in grayscale value compared to the control group was analyzed. Statistical analysis was performed using one-way ANOVA test. n=3. (**J**) TRAP staining of RAW264.7 cells with RANKL induction and TRAP-positive osteoclast number per well. Scale bars = 30 μm. (**K**) F-actin immunofluorescence staining of RANKL-stimulated RAW264.7-derived osteoclasts and multinucleated giant cell quantity per well. Scale bars = 25 μm. (**L, M**) RAW264.7 cells were stimulated by RANKL for 3 days. Western blot analysis of NFATc1, CTSK, and MMP9 was performed, and the fold change in grayscale value compared to the control group of western blot results was determined. n=3. Statistical analysis was performed using Student's t test. (**N**) TRAP staining of mouse knee joint subchondral bone tissue. Scale bars = 100 μm. (**O**) Osteoclasts as a percentage of the bone surface analysis of histological TRAP staining results. n=5. Data shown represent the mean ± SD. *p<0.05, **p<0.01, ns = no significance.

The online version of this article includes the following source data and figure supplement(s) for figure 3:

**Source data 1.** Uncropped western blot images for *Figure 3*.

**Source data 2.** Raw microscopy images for *Figure 3*.

**Figure supplement 1.** GANT58 at working concentration has no obvious apoptosis-promoting effect on cells.

the influence of drug-induced apoptosis on osteoclast activation, we carried out live/dead staining and apoptosis detection on BMMs. We found that there was no obvious death or apoptosis at a working concentration of 10 μM. However, when the concentration was increased by four times, more apoptotic cells appeared (*Figure 3—figure supplement 1A, B*). Osteoclast formation is accompanied by high expression of NFATc1. Immunofluorescence staining showed that the number of NFATc1-positive cells decreased significantly after GANT58 treatment (*Figure 3E*). We further performed the same treatment on the macrophage cell line RAW264.7. Tartrate-resistant acid phosphatase (TRAP) and F-actin staining confirmed that GANT58 intervention greatly reduced the formation of mature osteoclasts as well (*Figure 3F and G*). Then, we extracted the total protein from RAW264.7 cells with different treatments and performed western blot experiments. According to the results, we found that GANT58 hindered the activation of GLI1 translocation into the nucleus (*Figure 3H and I*). Subsequently, we further induced osteoclasts in WT and *Gli1*-knockdown macrophages. The results showed that the inhibition of GLI1 expression reduced the formation of osteoclasts, suggesting the specificity of GLI1 in regulating the activation of osteoclasts (*Figure 3J and K*). During osteoclast induction, both GANT58 treatment and *Gli1* knockdown significantly inhibited the expression of markers related to osteoclast function, including NFATc1, CTSK and MMP9 (*Figure 3L and M*). To further verify the effect of GLI1 osteoclast activation in RA, we performed TRAP staining of knee subchondral bone in mice of different groups. The results showed that there was more osteoclast formation around the bone tissue in the CIA model group. After GANT58 treatment, the number of osteoclasts decreased, and the trabecular morphology was close to the normal tissue level (*Figure 3N and O*). In summary, these data suggested that the activity of GLI1 may facilitate osteoclast formation.

## GLI1 regulates the expression of DNMTs in distinct ways during the different fates of macrophages

As a nuclear transcription factor, GLI1 exerts an active effect through nuclear entry. To explore the potential downstream regulatory mechanism of GLI1, RNA sequencing (RNA-seq) of macrophages in resting state before and after GLI1 intervention was performed to observe gene expression changes. The seq data showed that more genes were downregulated (143) than upregulated (74) in GANT58-treated cells (*Figure 4—figure supplement 1A, B*). Among these differentially altered genes, we revealed through Gene Ontology (GO) analysis that GANT58 intervention in GLI1 affected multiple biological processes, including macrophage chemotaxis and macrophage cytokine production (*Figure 4A*). Moreover, Kyoto Encyclopedia of Genes and Genomes (KEGG) enrichment analysis of the enriched pathways was then performed, and we identified the top 30 enriched pathways. In these pathways, we classified them into cellular processes (red), human diseases (blue) and organismal systems (green). These downregulated genes were involved in the development of human diseases such as rheumatoid arthritis, as well as organismal systems such as osteoclast differentiation (*Figure 4B*, *Figure 4—figure supplement 1C*). This evidence confirmed our previous results. Specifically, GANT58 reduced some of the osteoclast and inflammation-related genes in the cell resting state.

It is worth noting that among the genes with altered expression, the *Dnmt3a* gene of the DNA methyltransferase family showed a decreasing trend (*Figure 4C*). DNA methylation is an important

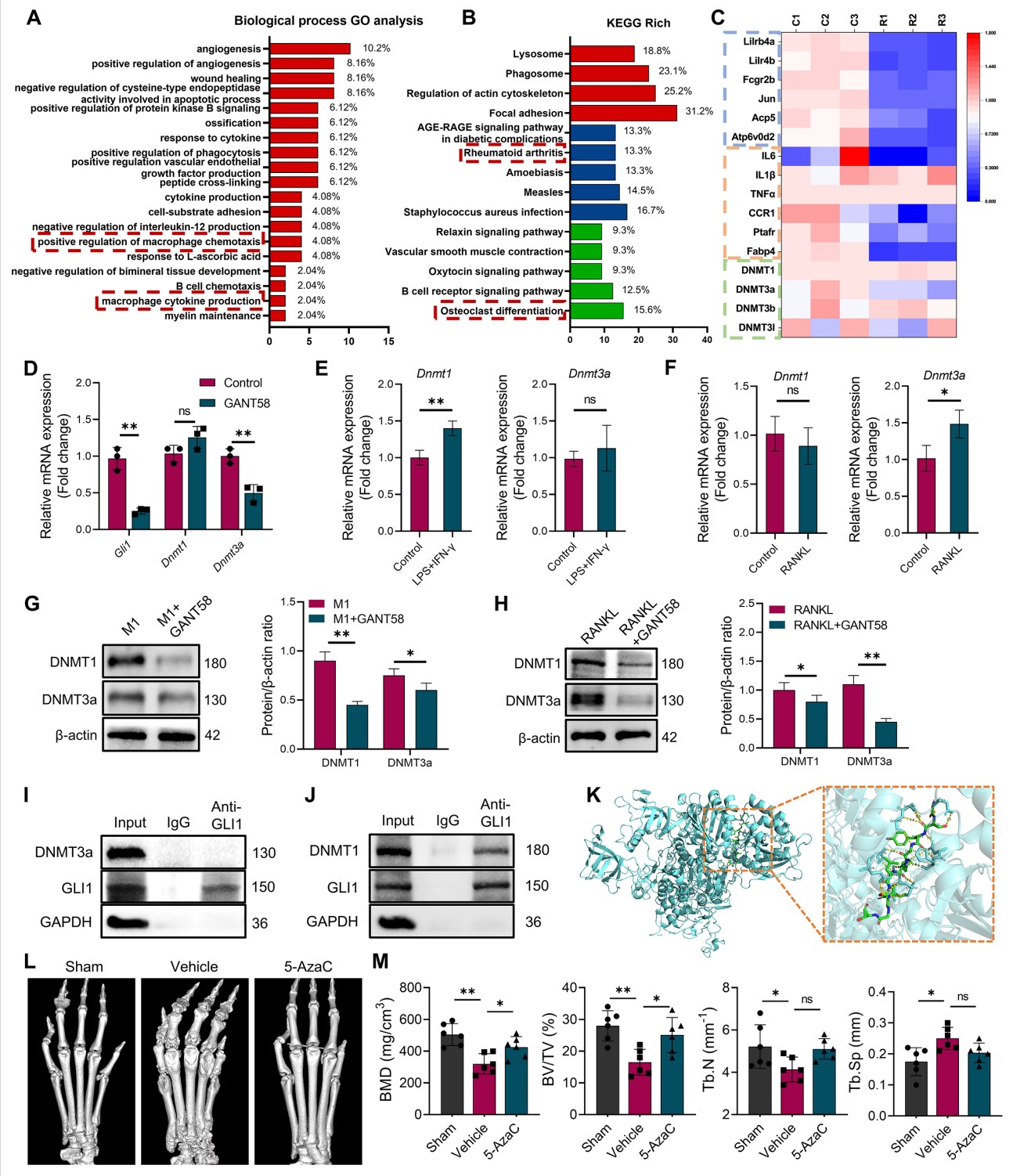

**Figure 4.** DNA methyltransferases might be a regulatory target downstream of GLI1. (**A**) Biological process GO analysis of RNA-seq results for macrophages with or without GANT58 treatment. (**B**) KEGG rich analysis of RNA-seq results. (**C**) Heat map of parts of the relevant gene transcriptional expressions (C=control group; R=GANT58-treated group; red: increased expression; blue: decreased expression). (**D**) Relative mRNA expression of *Gli1*, *Dnmt1* and *Dnmt3a* in macrophages with or without GANT58 treatment. Statistical analysis was performed using two-way ANOVA test. (**E**) RAW264.7 cells were stimulated by LPS and IFN-γ for 24 h, relative mRNA expression of *Dnmt1* and *Dnmt3a* was detected by RT-qPCR. (**F**) RAW264.7 cells were stimulated by RANKL for 48 hr, relative mRNA expression of *Dnmt1* and *Dnmt3a* was detected by RT-qPCR. n=3. Statistical analysis was performed using Student's t test. (**G**) RAW264.7 cells were stimulated by LPS and IFN-γ (M1 induction) for 24 hr, with or without GANT58 co-intervention. Western blot

*Figure 4 continued on next page*

*Figure 4 continued*

results of DNMT1 and DNMT3a protein expression and grayscale value ratio to β-actin of western blot results. n=3. (**H**) RAW264.7 cells were stimulated by RANKL for 3 days, with or without GANT58 co-intervention. Western blot results of DNMT1 and DNMT3a protein expression and grayscale value ratio to β-actin of western blot results. n=3. Statistical analysis was performed using two-way ANOVA test. (**I, J**) Co-IP detection of protein binding between GLI1 and DNMT1/DNMT3a. n=3. (**K**) Protein–protein interface interaction of GLI1 and DNMT1 with PyMOL. (**L**) Micro-CT scanning and 3D reconstruction of mouse paws. (**M**) Bone parameters of BMD, BV/TV, Tb.N, Tb.Th. n=6. Statistical analysis was performed using one-way ANOVA test. Data shown represent the mean ± SD. *p<0.05, **p<0.01, ns = no significance.

The online version of this article includes the following source data and figure supplement(s) for figure 4:

**Source data 1.** Uncropped western blot images for *Figure 4*.

**Source data 2.** RNA-seq data for *Figure 4*.

**Figure supplement 1.** Analysis of differentially expressed genes and enrichment pathways of RNA-seq results.

**Figure supplement 2.** DNMT1 and DNMT3a are highly expressed in synovial tissue of patients with rheumatoid arthritis.

**Figure supplement 2—source data 1.** Uncropped western blot images for *Figure 4—figure supplement 2*.

**Figure supplement 3.** GANT58 reduced the expression of DNMT1 and DNMT3a in macrophages under different induction conditions.

epigenetic modification that regulates gene expression and is activated by DNMTs (*Li and Zhang, 2014*). In mammals, DNMT1 and DNMT3a are the main methylation regulatory enzymes responsible for the de novo methylation and methylation maintenance of DNA. Furthermore, we searched for the correlation between GLI1 and DNMTs in mouse species on the public platform GeneMANIA. The results of bioinformatics analysis showed a latent relationship between GLI1 and DNMTs as well (*Figure 4—figure supplement 1D*). Thus, through RT–qPCR validation, we found that *Dnmt3a* was clearly inhibited after GANT58 intervention, but the expression of *Dnmt1* did not appear to change significantly (*Figure 4D*). Similar results were also observed in *Gli1* knockdown cells (*Figure 4—figure supplement 2A, B*). With further detection of DNMTs protein expression in human synovial tissues, we found that DNMT1 and DNMT3a were more highly expressed in RA patients (*Figure 4—figure supplement 2C, D*). Moreover, the RT–qPCR results showed that *Dnmt1* was more highly expressed in M1 macrophages, while *Dnmt3a* mRNA was more highly expressed during osteoclast induction (*Figure 4E and F*). To this end, we wondered whether there will be another outcome in different induction states. Therefore, we further examined DNMT1 and DNMT3a expression changes during M1 macrophage and osteoclast induction processes with or without GANT58 intervention. It is interesting to note that although GANT58 only affected the expression of the *Dnmt3a* gene in the resting state, the western blot results suggested that GANT58 reduced the expression of DNMT1 during M1 macrophage induction and the expression of DNMT3a during osteoclast induction (*Figure 4G and H*). The expression and localization of the proteins detected by immunofluorescence staining demonstrated that DNMT1 and DNMT3a were highly expressed in the nucleus under LPS/IFN-γ and RANKL stimulation, and GANT58 reduced their nuclear expression in both RAW264.7 cells and BMMs (*Figure 4—figure supplement 3A, B*). In view of the above findings, Co-IP protein binding experiments of cells in the resting states have found that DNMT1, but not DNMT3a protein, could be pulled down by the antibody of GLI1, suggesting that there was direct binding between GLI1 and DNMT1 (*Figure 4I and J*). Accordingly, we used the ZDOCK server to find possible binding sites and performed docking of the protein–protein interface interaction of GLI1 and DNMT1 with PyMOL (*Figure 4K*, *Supplementary file 2*). Overall, the above validation results confirmed that GLI1 might have a regulatory effect on DNMTs during different induction processes of macrophages, and this regulation mode was not exactly the same.

Based on the above results, we treated CIA mice with 5-azacytidine (5-AzaC), an inhibitor of DNMTs, as a therapeutic intervention to observe the potential regulatory effect of DNMTs on RA. As shown in *Figure 4L and M*, 5-AzaC significantly alleviated joint bone destruction in CIA mice and improved bone parameters. This suggested that DNMTs might be involved in the RA development process, further suggesting their potential regulatory relationship with GLI1.

## GLI1 regulates the proinflammatory phenotype of macrophages by affecting the expression of DNMT1

As an important member of the DNMT family, apart from being involved in the occurrence of tumor diseases, DNMT1 has also been shown to be associated with certain inflammatory diseases (*Wang*

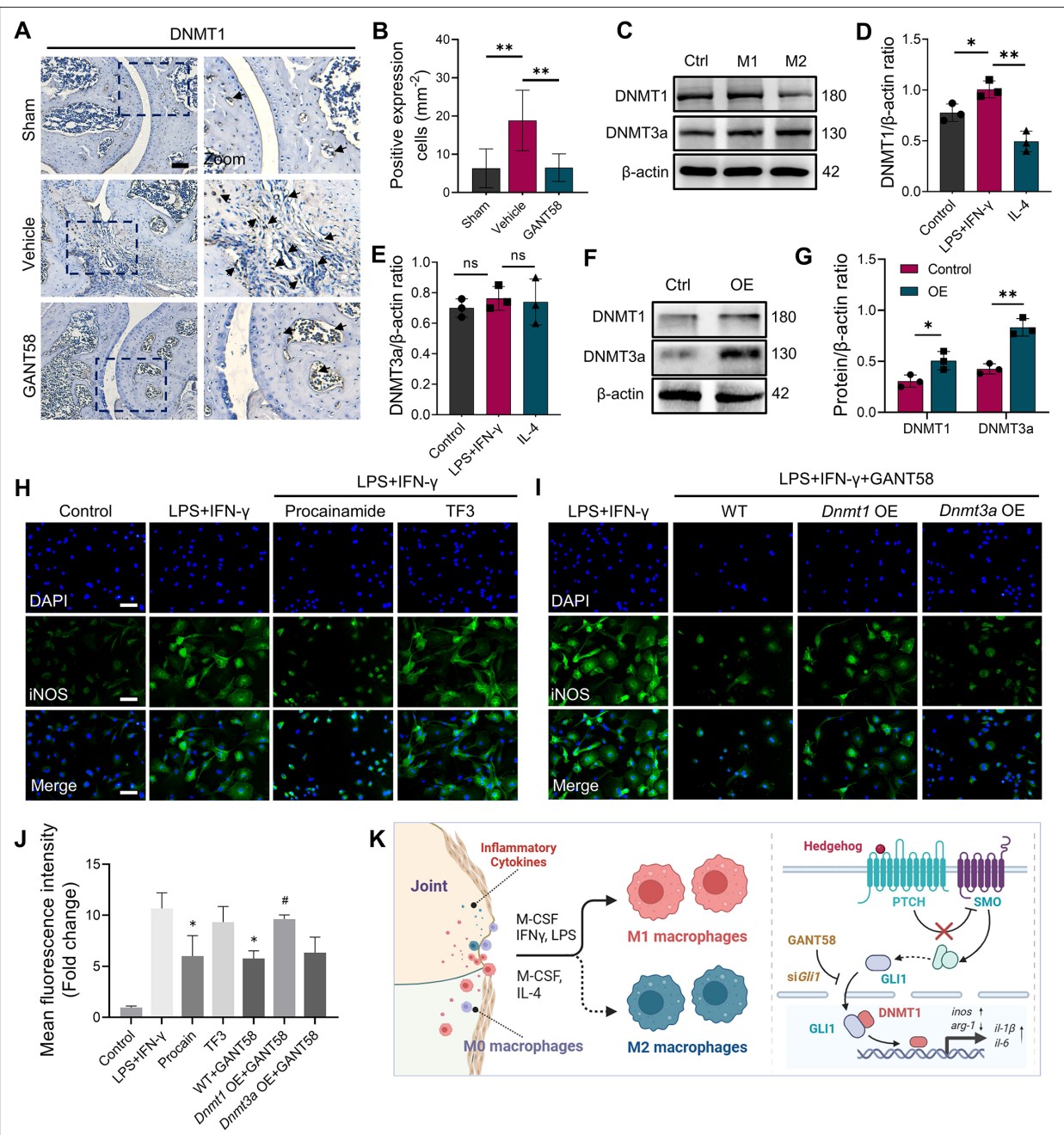

**Figure 5.** GLI1 regulates the expression of DNMT1, which affects macrophage phenotypes and the release of inflammatory cytokines. (**A**) IHC staining of DNMT1 and (**B**) quantification of positive cell numbers in mouse ankle joints. n=5. Scale bars = 100 μm. Statistical analysis was performed using one-way ANOVA. (**C**) RAW264.7 cells were stimulated by LPS/IFN-γ and IL-4 for M1 and M2 polarization. Western blot detection of DNMT protein expression at 24 h. (**D**) DNMT1 grayscale value ratio to β-actin of western blot results. n=3. *p<0.05. (**E**) DNMT3a grayscale value ratio to β-actin of western blot results. n=3. ns = no significance. (**F**) RAW264.7 cells were transfected with *Dnmt3a* and *Dnmt1* overexpression lentiviruses. Western blot analysis of DNMT1 and DNMT3a protein expression. (**G**) Grayscale value ratio to β-actin of western blot results. n=3. Statistical analysis was performed using Student's t test. *p<0.05, **p<0.01. (**H, I**) BMMs were cultured and stimulated with LPS/IFN-γ in the presence of different interventions. Immunofluorescence staining of iNOS in LPS/IFN-γ-induced BMMs and (**J**) relative quantitative analysis of mean fluorescence intensity. Scale bars = 10 μm. n=3. (**K**) Schematic diagram of the regulatory mechanism. Statistical analysis was performed using one-way ANOVA test. Data shown represent the mean ± SD. *p<0.05 compared with the LPS/IFN-γ group; #p<0.05 compared with the WT+GANT58 group.

The online version of this article includes the following source data and figure supplement(s) for figure 5:

**Source data 1.** Uncropped western blot images for *Figure 5*.

**Figure supplement 1.** GANT58 decreased the expressions of DNMT1 in CIA mice.

*Figure 5 continued on next page*

*et al., 2016*). By staining mouse joints for IHC, we found that DNMT1 was more highly expressed in the CIA model group, while its expression was reduced in the GANT58 treatment group (*Figure 5A and B*, *Figure 5—figure supplement 1A, B*). We first observed the alteration of DNMTs during in vitro macrophage polarization induction. The western blot results showed that DNMT1 was more highly expressed during M1 macrophage induction (stimulated by LPS and IFN-γ) than during M2 induction (stimulated by IL-4; *Figure 5C–E*). To investigate the effect of DNMTs on macrophages, the DNMT3a-specific inhibitor theaflavin-3,3'-digallate (TF3) and the DNMT1-specific inhibitor procainamide (PR) were utilized for DNMTs intervention. The CCK-8 results showed that TF3 did not inhibit cell proliferation at concentrations ranging from 0 to 5 μM in either BMMs or RAW264.7 cells, and no obvious proliferative toxic effect of PR could be seen on BMMs or RAW264.7 cells (*Figure 5—figure supplement 2A–D*). We used different interventions to treat LPS/IFN-γ-induced macrophages and performed immunofluorescence staining to observe the expression of iNOS in proinflammatory M1 macrophages. An increase in the number of iNOS-positive cells was observed after stimulation with LPS and IFN-γ, and similar to the effect of GANT58, the inhibition of DNMT1, but not DNMT3a, was able to reduce M1 macrophage numbers (*Figure 5H*). In addition to pharmacological intervention, we also knocked down *Dnmts* using siRNA (*Figure 5—figure supplement 2E, F*). Immunofluorescence staining results confirmed that the expression of iNOS was decreased in si*Dnmt1* cells compared with WT cells after stimulation by LPS/IFN-γ (*Figure 5—figure supplement 3A*). According to this result, to demonstrate the downstream regulatory effect, we overexpressed (OE) *Dnmt1* and *Dnmt3a* by lentiviral transfection (GenePharma, Suzhou, China; *Figure 5F and G*). When LPS/IFN-γ was applied to *Dnmt1* OE macrophages, compared with WT cells, it seemed that there was no obvious effect on iNOS expression (*Figure 5—figure supplement 4A–C*). However, the results showed that the inhibitory effect of GANT58 on M1 macrophages was reversed by *Dnmt1* overexpression (*Figure 5H–J*). In addition, we measured the mRNA expression of related cytokines, including the proinflammatory cytokines *Il1b*, *Il6*, and *Tnfa* and the anti-inflammatory cytokine *Il10*, and found the same trend as that of the fluorescence staining results (*Figure 5—figure supplement 3B–E*). These findings suggest that the regulatory effects of GLI1 on proinflammatory cells are mediated by DNMT1 (*Figure 5K*).

## Overexpression of DNMT3a reverses GLI1 inhibition-induced osteoclast-forming disorders

IHC staining of tissue specimens showed that similar to the GLI1 expression trend, DNMT3a expression was elevated in the CIA model group, which could be decreased by GANT58 (*Figure 6—figure supplement 1A–C*). Meanwhile, western blot results showed that DNMT3a was also highly expressed during RANKL induction, while DNMT1 had no obvious change (*Figure 6A and B*). To investigate the effect of DNMT3a and DNMT1 on osteoclast formation, the DNMT3a-specific inhibitor TF3 and the DNMT1-specific inhibitor PR were used for intervention during osteoclast induction. We first treated both primary BMMs and RAW264.7 cells with TF3 and PR simultaneously during osteoclast induction. As a result, RANKL-induced osteoclast formation was strongly inhibited by TF3 but not PR treatment (*Figure 6D–G*). IRF8, a negative regulator of osteoclast activation, is downregulated by DNA methylation during osteoclast formation. Through RT–qPCR detection, it was found that the expression of *Irf8* in si*Dnmt3a* cells was higher than that in the RANKL control group (*Figure 6—figure supplement 1D*). In addition, osteoclast formation was inhibited in si*Dnmt3a* cells compared with WT RAW264.7 cells after induction by RANKL (*Figure 6H and I*). Notably, when *Dnmt3a* OE RAW264.7 cells were induced to differentiate into osteoclasts, we found that although there was no significant difference in the number of osteoclasts compared with the WT group at the end, more osteoclasts appeared in the *Dnmt3a* OE group on the third day of induction, suggesting that *Dnmt3a* OE may accelerate the activation of osteoclasts to some extent (*Figure 6—figure supplement 2A, B*). To verify whether the regulatory effect of GLI1 on osteoclasts was related to DNMTs, we pretreated WT RAW264.7 cells and

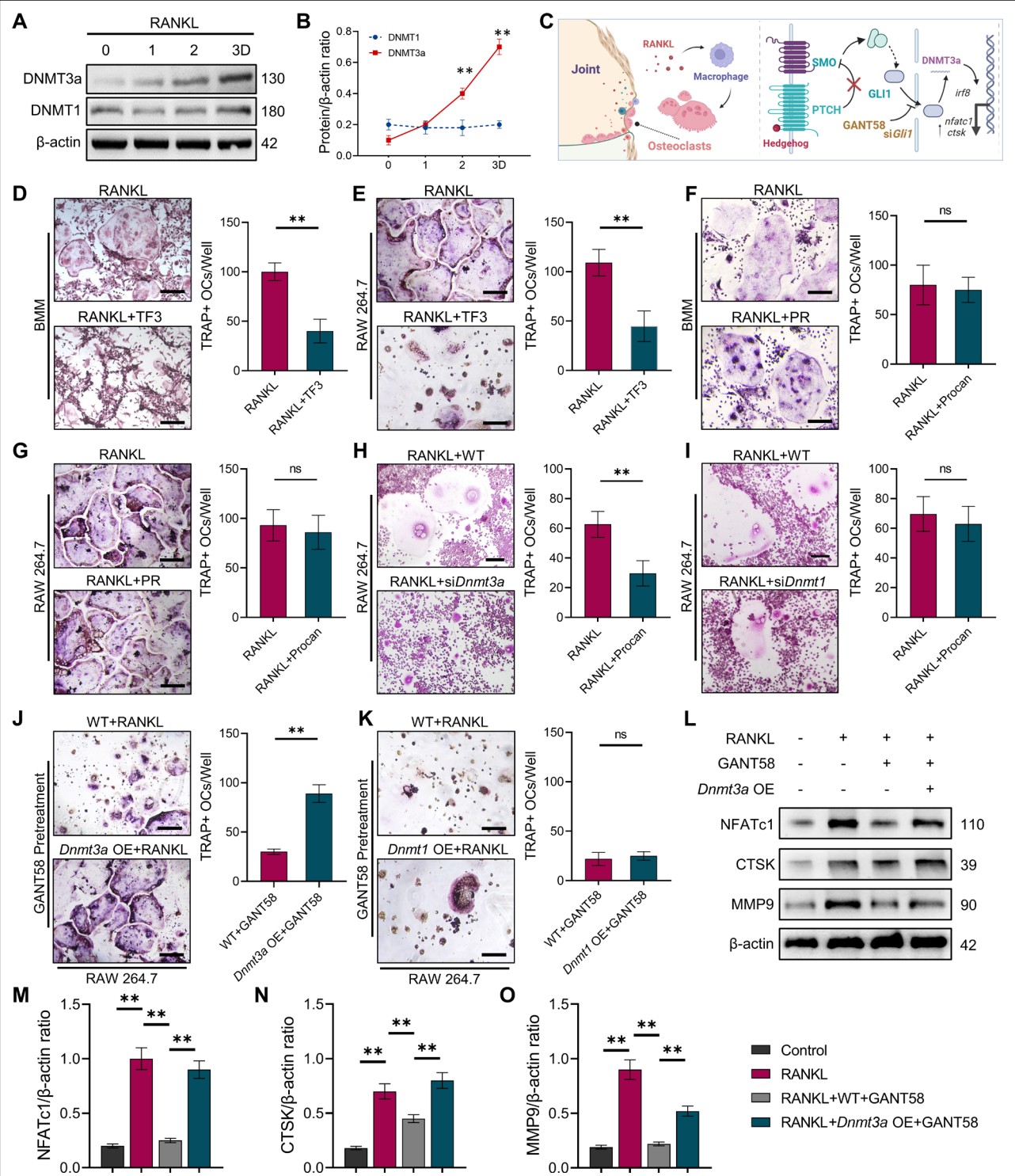

**Figure 6.** GLI1 regulates DNMT3a to affect the differentiation of osteoclasts. (**A**) RAW264.7 cells were stimulated with RANKL for 3 days. Western blot analysis of DNMT3a and DNMT1 during the process of osteoclast induction and (**B**) grayscale value ratio to β-actin of western blot results. n=3. (**C**) Schematic diagram of the regulatory mechanism. (**D, E**) TRAP staining of BMMs and RAW264.7 cells stimulated by RANKL (50 ng/ml) in the presence or absence of TF3 (2 μM) and TRAP-positive osteoclast quantity per well. Scale bars = 25 μm. (**F, G**) TRAP staining of BMMs and RAW264.7 cells stimulated by RANKL (50 ng/ml) in the presence or absence of procainamide (PR, 10 μM) and TRAP-positive osteoclast quantity per well. Scale bars = 25 μm. (**H**) WT and si*Dnmt3a* treated RAW264.7 cells were stimulated by RANKL. TRAP staining and TRAP-positive cell quantity per well. Scale bars = 25 μm. (**I**) WT and si*Dnmt1* treated RAW264.7 cells were stimulated by RANKL. TRAP staining and TRAP-positive cell quantity per well. Scale bars = 25 μm. (**J**) WT and *Dnmt3a* OE RAW264.7 cells were stimulated by RANKL in the presence of GANT58 intervention. TRAP staining and TRAP-positive

*Figure 6 continued on next page*

*Figure 6 continued*

cell quantity per well. Scale bars = 25 μm. (**K**) WT and *Dnmt1* OE RAW264.7 cells were stimulated by RANKL in the presence of GANT58. TRAP staining and TRAP-positive cell quantity per well. Scale bars = 25 μm. Statistical analysis was performed using Student's t test. (**L**) Western blot results of NFATc1, CTSK and MMP9 protein expression. (**M–O**) Grayscale value ratio to β-actin of western blot results. n=3. Statistical analysis was performed using one-way ANOVA test. Data shown represent the mean ± SD. *p<0.05, **p<0.01, ns = not significant.

The online version of this article includes the following source data and figure supplement(s) for figure 6:

**Source data 1.** Uncropped western blot images for *Figure 6*.

**Figure supplement 1.** GANT58 decreased the expressions of DNMT3a in CIA mice.

**Figure supplement 2.** Effects of overexpression of DNMT3a on induction of osteoclasts.

---

*Dnmt3a*- and *Dnmt1*-overexpressed RAW264.7 cells with GANT58 for 6 hr and then induced these cells with RANKL. After RANKL induction, we found that the number of osteoclasts among *Dnmt3a* OE cells was more increased than that in WT cells (*Figure 6J*), while in *Dnmt1* OE cells, osteoclast formation was still inhibited (*Figure 6K*). We analyzed the osteoclast formation- and function-related proteins NFATc1, CTSK, and MMP9 and found that GANT58 reduced the expression of these proteins. However, these proteins were upregulated when DNMT3a was overexpressed (*Figure 6L–O*). These findings suggested that DNMT3a might be downstream of GLI1 during the regulation of osteoclast formation (*Figure 6C*).

## Discussion

In the development of RA, synovitis infiltration and osteoclast overactivation are the direct causes of joint damage (*Firestein, 2003*). To find a better treatment strategy, we hope to identify a key target that can regulate inflammatory bone destruction using its specific inhibitor and clarify its specific mechanism to carry out targeted interventions and achieve the desired therapeutic effect. Macrophages are an important source of inflammatory cytokines and the primary cells associated with osteoclast formation (*Shapouri-Moghaddam et al., 2018*). In RA, M1 macrophage polarization is generally considered an important factor that promotes the development of inflammation (*Matsumoto et al., 2016*). The release of the proinflammatory cytokines IL-1β, IL-6, and TNF-α promotes the activation of more osteoclasts and leads to the destruction of bone structure.

GLI1 is a downstream transcription factor of the hedgehog-GLI signaling pathway. After hedgehog is activated, GLI1 and its factors form a complex with microtubules, enter the nucleus and activate downstream gene transcription (*Kopinke et al., 2021*). In previous studies, GLI1 signal transduction and other pathways, including the NF-κB signaling pathway, were usually studied in tumor-associated diseases and are considered a response network that promotes cancer development (*Wils and Bijlsma, 2018*; *Zeng and Ju, 2018*). Qin. et al. found that the content of SHH in RA patient serum increased significantly compared with that in healthy patients (*Qin et al., 2016*). At the same time, our study also showed that GLI1 was more highly expressed in the joint tissue of RA patients. These results suggest that the HH-GLI signaling pathway may be involved in the regulation of the pathological process of RA. However, the research results of the hedgehog pathway in bone metabolism are complex. In Heller's report, enhanced hedgehog signaling due to PTCH heterozygosity indirectly enhanced tumor-induced bone resorption (*Heller et al., 2012*); Ling also demonstrated that the inhibition of GLI1 maintained bone mass (*Ye et al., 2018*). These results showed that GLI1 had a potential connection with the formation of osteoclasts. In contrast, Shoko Onodera concluded that impaired osteoblastogenesis was restored to normal levels by treatment with a small molecule that actives hedgehog signaling (*Onodera et al., 2020*). This diversity may be due to the unknown complex mechanism of various factors in the activation of the hedgehog signaling pathway. In our study, we found that GANT58 had an obvious inhibitory effect on the osteoclastic induction of BMMs and RAW264.7 cells, which was consistent with the results of Heller's study. This seems to mean that the activation of GLI1 may have multiple effects on the regulation of bone tissue homeostasis and maintain a balance in the physiological state. Inflammation is an important reason for osteoclast activation (*Hwang et al., 2018*). The release of proinflammatory cytokines promotes arthritis and osteoclastogenesis (*Tran et al., 2018*; *Zhu et al., 2018*). Our study showed that GANT58 inhibited the formation of M1 macrophages in an inflammatory environment and inhibited the expression of the

proinflammatory cytokines IL-1β, IL-6, and TNF-α. Interestingly, in previous studies, it was found that GLI1 did not appear to respond very strongly to LPS intervention alone (*Collins et al., 2012*; *Sheng et al., 2021*; *Zhang et al., 2022*). In this study, GLI1 played a positive role in the formation of M1 macrophages, and at this point, the role of IFN-γ may be more important. Studies have reported that simple IFN-γ stimulation promotes the expression of GLI1 in neuron precursor cells (*Sun et al., 2010*; *Wang et al., 2021*). These results seem to further confirm that the immunomodulatory role of GLI1 in RA is mainly achieved by macrophage polarization changes. In subsequent studies, we may be able to further explore the regulatory effects of LPS and IFN-γ on GLI1.

To clarify the downstream regulatory mechanism of GLI1, bioinformatics prediction analysis and RNA-seq were performed and ultimately identified the downstream DNMT family. In addition to normal physiological development, the abnormal expression of DNMTs causes the development of tumors and other diseases (*Mohd Murshid et al., 2022*). Through treatment with DNMT inhibitors, inflammatory arthritis in mice was significantly relieved, which was consistent with previous studies (*Tóth et al., 2019*). These results suggested that DNMTs might be involved in the inflammatory reaction and bone destruction of RA. Reports have suggested that the absence of DNMT3a inhibits the formation of osteoclasts, which may be due to the methylation of downstream IRF8 by DNMT3a (*Nishikawa et al., 2015*). In our study, we also verified this finding through pharmacological and genetic interventions. Similarly, in myeloma disease models, high expression of DNMT3a is thought to promote hypermethylation of Runx2, osterix and IRF8 CpG islands and inhibit the expression of these genes, thus inhibiting osteoblasts and promoting osteoclast activation (*Wen et al., 2020*). In contrast to DNMT3a, the overexpression of *Dnmt1* promotes proinflammatory cytokine production in macrophages and plasma during atherosclerosis and inflammation (*Anto Michel et al., 2018*; *Yu et al., 2016*). Notably, GANT58 significantly inhibited DNMT1 and DNMT3a activation during M1 macrophage and osteoclast induction, and overexpression of *Dnmt3a* and *Dnmt1* reversed the inhibitory effect of GANT58 on the formation of osteoclasts and M1 macrophages. According to the sequencing and research results, GLI1 activity is closely related to the expression activity of DNMTs. In the cell resting state, we found that the inhibitory effect of GANT58 on DNMTs was limited to DNMT3a at the level of gene transcription, while Co-IP analysis found a binding relationship between GLI1 and DNMT1 but not DNMT3a. This evidence suggested that the active regulation of GLI1 on DNMT1 might be post-protein translation, while the regulation of DNMT3a expression might be located at the gene transcriptional level. The molecular function of GLI1 usually acts as a transcriptional activator. Patricia González Rodríguez's latest research shows that during autophagy induction, selective GLI1 enrichment can be observed in the regions closer to the transcription start site (TSS) of the DNMT3a gene (*González-Rodríguez et al., 2022*). This finding supports our speculation that GLI1 may play a regulatory role by combining with the DNMT3a promoter region sequence.

In vivo experiments using the GLI1 inhibitor GANT58 showed its anti-inflammatory role and protection of bone mass. GANT58 showed good therapeutic effects on CIA mice and inhibited the number of osteoclasts around bone tissue. These results not only elucidated, to a certain extent, the molecular mechanism by which GLI1 influences the pathological process of RA but also showed the potential therapeutic effect of GANT58 on inflammatory bone destruction in RA, which might have clinically translatable research implications. Although we have demonstrated that the inhibition of GLI1 by GANT58 can reduce the inflammatory response and inhibit osteoclast formation and that this mechanism is achieved through the downregulation of DNMTs, these findings also raise new questions. In a previous research report, *Gli1* haplodeficiency in mice decreased bone mass with reduced bone formation compared to control mice, which was due to osteoblasts with weakened function (*Kitaura et al., 2014*). In this process, the osteogenic differentiation of mesenchymal stem cells also affects the function of osteoclasts. In addition, GLI1 is also used as a marker for osteogenic progenitors, which are precursors of chondrocytes and osteoblasts (*Shi et al., 2017*). These studies suggest that the regulation of GLI1 on bone metabolism is complex, and the therapeutic effect of GANT58 on RA may be more than just affecting the inflammatory reaction mediated by macrophages and the bone destruction mediated by osteoclasts. In addition to macrophages and osteoclasts, synovial fibroblasts and osteoblasts play essential roles in the RA microenvironment. These cells are also closely linked to each other. Synovial fibroblast OPG and RANKL secreted by osteoblasts are important factors that regulate osteoclasts. Therefore, in a follow-up study, we will extend the study of GLI1 to its regulatory mechanism in osteoblasts.

## Materials and methods

### Experimental animals and human synovial tissue

All animals were ordered through the Animal Experiment Center of Soochow University. Male DBA mice aged 6–8 weeks and weighing 15–20 $g$ were randomly selected and fed in a specific pathogen-free (SPF) environment at a room temperature of 25 °C, a relative humidity of 60%, and 12 hours of alternating light. All animal experiments were approved by the Animal Ethics Committee of Soochow University (201910 A354). The animals were divided randomly into groups (6 per group): sham group (healthy mice not received any treatment), vehicle control group (CIA model mice treated with solvent), and GANT58 (GLI1-specific inhibitor; MedChemExpress, New Jersey, USA) group (mice treated with 20 mg/kg GANT58) or 5-AzaC (DNMT-specific inhibitor; MedChemExpress) group (mice treated with 2 mg/kg 5-AzaC). An emulsion of bovine type II collagen (Chondrex, Redmond, WA, USA) and an equal amount (1:1, v/v) of complete Freund's adjuvant (Chondrex) were prepared to establish the CIA mouse model. First, 0.1 ml of the emulsion was injected intradermally into the base of the tail on day 0. On day 21, 0.1 mg of bovine type II collagen mixed with incomplete Freund's adjuvant (Chondrex) was injected. For the vehicle group, mice were injected with the same volume of placebo daily. For the treatment groups, mice were injected with GANT58 or 5-AzaC solution daily. All interventions began the day after the second injection of bovine type II collagen. The arthritis score was given every three days from the second immunization. On day 49, all mice were sacrificed (in accordance with the guidelines of the Animal Welfare and Ethics Committee of Soochow University) for the collection of specimens. The severity of arthritis was scored every 3 day independently by three researchers who were blinded to the order in which the mice were presented. The paws of all of the mice were assessed and scored from 0 to 4 (0: normal; 1: slight swelling of the joint; 2: obvious local swelling of the joint; 3: extensive swelling of the joint with limited movement; and 4: extensive swelling of the joint and inability to bend the joint) according to the degree of joint swelling and mobility limitation. Normal human synovial tissues were donated from surgical patients who had noninflammatory knee joint diseases such as injury in traffic accidents or traumatic fractures. RA synovial tissues were obtained from RA patients. All included patients were scored DAS28 as described (*Vadell et al., 2020*). The samples were obtained with the informed consent of the patients, and all operations were approved by the Ethics Committee of the First Affiliated Hospital of Soochow University ((2018) Ethical Approval No.012).

### Immunohistochemical staining

Tissue samples were dewaxed and rehydrated gradually with xylene and gradient alcohol, and prethen citrate buffer solution was added for antigen repair. After the samples were washed with PBS, serum was added to the tissue surface and incubated for 30 min at room temperature. The required primary antibody dilutions (GLI1, 1:200; DNMT1, 1:200; DNMT3a, 1:200; Abcam, Cambridge, UK) were prepared, added to the samples after the serum was removed, and incubated overnight at 4 °C. After the samples were washed with PBS, biotin-labeled secondary and antibody working solutions (VECTOR, Burlingame, CA, USA) were gradually added to the tissue samples and incubated for 30 min separately. A controlled diaminobenzidine (DAB; Cell Signaling Technology, Danvers, MA, USA) color reaction was performed under a microscope, and the reaction was stopped with distilled water immediately after color development. Hematoxylin staining was then performed. Finally, the samples were immersed in an 80% ethanol solution, a 95% ethanol solution and anhydrous ethanol successively for dehydration. Images were visualized by optical microscopy, and positively stained cells were assessed by ImageJ software (Version 1.8.0.112).

### Micro-CT analysis

Fixed bone samples of mice were collected. The joint samples were placed in a SkyScan 1174 Micro-CT scanning warehouse (Belgium). The parameters were set as follows: voltage 50 kV, current 800 μA, scanning range 2 cm × 2 cm, and scanning layer thickness 8 μm. The scan data were then entered into a computer to conduct three-dimensional reconstruction with NRecon software (Bruker, Germany), and the bone tissue parameters were analyzed with CTAn software (Bruker, Germany) after data conversion. During this procedure, we performed an analysis of bone parameters, including BMD (bone mineral density), BV/TV (percentage trabecular area), Tb. N (trabecular number) and Tb. Sp (trabecular separation) by selecting the small joint of the paws as the region of interest (ROI, bone

tissue from ankle joint to toe) in CTAn software. The three-dimensional reconstruction images were exhibited by Mimics Research software (Version 21.0; Materialise, Belgium).

## Hematoxylin and eosin (H&E) staining

The fixed murine bone tissues were removed, placed in 10% ethylenediaminetetraacetic acid (EDTA, Sigma, St. Louis, Missouri, USA) for 3 weeks and paraffin-embedded and sectioned. Other tissues were paraffin embedded and sectioned after being fixed directly. The tissue sections were placed into xylene to dissolve the wax. The sections were then rehydrated in an ethanol solution. After being washed, the sections were immersed in hematoxylin dye (Leagene, Beijing, China) for 3 min. Color separation with 1% hydrochloric acid in ethanol and ammonia was then performed. The sections were then immersed in eosin dye (Leagene) and dehydrated with ethanol and xylene after being washed. Bone erosion and fibrosis were visualized and assessed by optical microscopy. Inflammatory cell infiltration was scored from 0 to 3 (0: no infiltration; 1: a small amount of local infiltration; 2: extensive local infiltration; 3: extensive infiltration into the joint capsule with the formation of agglomerates) according to H&E staining.

## Cell culture

Murine BMMs and RAW264.7 cells were used in this experiment. BMMs were extracted from the C57BL/6 mouse long bones of the lower limbs and cultured in α-minimum essential medium (MEM; HyClone, California, USA) containing 10% fetal bovine serum (FBS; Gibco, California, USA), 30 ng/ml macrophage colony stimulating factor (M-CSF; R&D Systems, Minnesota, USA) and 100 U/ml penicillin/streptomycin (P/S, NCM Biotech, Suzhou, China). RAW264.7 (National Collection of Authenticated Cell Cultures; TCM13) cells were cultured in DMEM (HyClone). All cells were cultured in a 37 °C standard environment and stored at –80 °C with serum-free cell freezing medium (NCM Biotech, Suzhou, China).

## Cell viability

Cell viability was assessed by a cell counting kit 8 (CCK-8; ApexBio, Houston, USA) assay according to the manufacturer's protocol. The inhibition rate was calculated as follows: inhibition rate (x) = ($OD_{control}$ - $OD_x$)/$OD_{control.}$ A Live/Dead cell staining kit (Yeasen) consisting of Calcein-AM (green fluorescence) and propidium iodide (PI, red fluorescence) and Annexin V/PI staining kit (Beyotime, Shanghai, China) was used according to the instructions to assess cell apoptosis.

## Tartrate-resistant acid phosphatase (TRAP) staining

A TRAP staining kit (Sigma) was used in this experiment. The dye solution was prepared according to the instructions. For tissue staining, after the tissue sections were dewaxed, a low concentration ethanol solution was added to rehydrate the sections. The repair solution was added to the surface of the tissue for antigen repair. Then, TRAP solution was added and incubated in the dark for 1 hr. For osteoclast induction, BMMs and RAW264.7 cells were seeded in 24-well plates at densities of 4×104/well and 6×104/well, respectively. RANKL (50 ng/ml) was added to the medium. For cell staining, the cells were first fixed with 4% paraformaldehyde for 15 min and then incubated with TRAP dye solution for 40 min. The positive cells were observed under an inverted optical microscope.

## Immunofluorescence staining

BMMs were seeded in 24-well plates at a density of 4×10⁴/well and stimulated with RANKL (50 ng/ml) or LPS (100 ng/ml)+IFN-γ (20 ng/ml). After culturing and stimulation, all cells were fixed with 4% paraformaldehyde for 15 min. To fully bind the antibody to the antigen, 0.1% Triton X-100 (Beyotime) solution was added and incubated on ice for 10 min. Then, the cells were washed with PBS 3 times, and blocking buffer was added to the cells for 1 hr. The cell supernatant was removed, and the prepared primary antibody solution was added, followed by incubation at 4 °C for 12 hr. The primary antibody was removed, and the fluorescent secondary antibody solution was added and incubated at room temperature for 1 hr. Finally, DAPI dye (1:20; Yuanye, Shanghai, China) solution was added and incubated for 10 min, and then the cells were photographed under a fluorescence microscope (Leica, Wetzlar, Germany). Finally, the mean fluorescence intensity and co-location coefficient (Pearson's R value) was acquired and analyzed by ImageJ software (Version 1.8.0.112).

## ELISA

LPS and IFN-γ-induced cell supernatants were collected. The concentrations of the cytokines IL-1β, IL-6, TNF-α and IL-10 in the cell supernatants were measured by ELISA kits (Multi Sciences, Hangzhou, China). The experiments were performed in accordance with the protocol.

## Quantitative real-time polymerase chain reaction (PCR)

Total mRNA was extracted using Beyozol reagent (Beyotime). The concentration and purity of the mRNA were assessed by a Nanodrop spectrophotometer. The isolated mRNA (2 μg) was used for reverse transcription PCR to produce cDNA with an iScript cDNA synthesis kit (Bio-Rad). A total of 2 μL of cDNA product was used for subsequent RT–qPCR analysis using SYBR1 Premix Ex Taq (Takara, Dalian, Japan). All primers used in this study are shown in *Supplementary file 3*.

## Western blotting

Cells were seeded in 6-well plates at a density of $1 \times 10^6$/well with stimulation with RANKL (50 ng/ml) or LPS (100 ng/ml)+IFN-γ (20 ng/ml) or IL-4 (20 ng/ml). First, cells were collected to extract total protein, and the BCA (Beyotime) method was used to adjust the protein concentration. Total protein was mixed with 5×loading buffer (Beyotime) and boiled at 95 °C for 10 min. For cytoplasmic/nuclear isolation, cells were collected, and protein was extracted according to the instructions using a nuclear protein and cytoplasmic protein extraction kit (Beyotime). The proteins were separated by SDS polyacrylamide gel electrophoresis (SDS–PAGE; EpiZyme, Shanghai, China) based on their different molecular weights. Electrophoresis was performed using Bio-Rad (California, USA) equipment at 180 V for 40 min. Then, the proteins were transferred to a nitrocellulose membrane at 350 mA for 70 min using membrane transfer equipment (Bio-Rad). The membrane was removed and placed into western blot blocking buffer for 1 hr at room temperature. The diluted primary antibodies (GLI1, Abclonal, A14675; β-actin, Beyotime, AF5003; Lamin-B1, Abcam, ab16048; NFATc1, Abclonal, A1539; CTSK, Abclonal, A5871; MMP9, Abclonal, A11147; DNMT1, Abclonal, A16729; DNMT3a, Cell Signaling Technology, D23G1; GAPDH, Abclonal, A19056) were placed on the membrane and incubated at 4 °C for 12 hours, and then the corresponding secondary antibody was added and incubated for 1 hour at room temperature. Finally, a chemiluminescence detection system (Bio-Rad) was used to observe the results.

## High-throughput sequencing (RNA-seq)

To further screen for differentially expressed genes, we first subjected RAW264.7 cells to a 24 hr adaptive culture, followed by the addition of GANT58 at a final concentration of 10 μM to the GANT58 intervention group and cultured for a total of 24 hr. After the cell treatment was completed, cells of the control group and GANT58-treated group were collected, and RNA-seq detection and analysis were entrusted to a professional biological company (Azenta Life Sciences, Suzhou, China). Briefly, for differential expression gene analysis, the differential expression conditions were set as fold change (FC) >1.5 and false discovery rate (FDR)<0.05. Among the differentially expressed genes, we performed GO analysis of the biological process and showed the top 20 enriched biological activities. KEGG analysis of the pathway team enrichment was then performed, and we showed the top 30 enriched pathways. In these pathways, we classified them into cellular processes (red), human diseases (blue) and organismal systems (green) and showed the enrichment of TOP5 in each category.

## Coimmunoprecipitation (Co-IP)

After culture under the specified conditions, the cells to be tested were collected. Precooled RIPA buffer containing protease inhibitor was added and cracked at 4 °C for 30 min. Then, the samples were centrifuged for 15 min at 14,000 $g$, and the supernatant was immediately transferred to a new centrifuge tube. Protein A agarose was prepared, and the beads were washed twice with PBS and then prepared with PBS to 50% concentration. Protein A agarose beads (100 μL) were added per 1 ml of total protein (50%) and shaken at 4 °C for 10 min. After centrifugation at 4 °C and 14000×$g$ for 15 min, the supernatant was transferred to a new centrifuge tube, and protein A beads were removed. A certain volume of antibody (GLI1, Santa Cruz Biotechnology, sc-515751) was added to 500 μL of total protein, and the antigen antibody mixture was slowly shaken at 4 °C overnight. Protein A agarose beads (100 μL) were added to capture antigen-antibody complexes, and the antigen-antibody

mixture was slowly shaken overnight at 4 °C. After centrifugation at 14,000 rpm for 5 s, the agarose bead antigen antibody complex was collected, and the supernatant was removed and washed with precooled RIPA buffer hree times. The sample was boiled for 5 min, the supernatant was electrophoresed, and the remaining beads were collected. Protein blot analysis was carried out with DNMT1 (A16729) and DNMT3a (D23G1) antibodies.

## Statistical analysis

All data are presented as the mean ± standard deviation (SD). Statistical analysis was performed with an unpaired two-tailed Student's t test for single comparisons with GraphPad Prism 8 (GraphPad Software, CA, USA). One-way analysis of variance (ANOVA) was used to compare data from more than two groups. Bonferroni correction was used with one-way ANOVA for multiple comparisons. $p$ values less than 0.05 were considered statistically significant.

## Acknowledgements

This work was supported by the National Nature Science Foundation of China, the Natural Science Foundation of Jiangsu Province, the Priority Academic Program Development of Jiangsu Higher Education Institutions (PAPD), the Science and Technology Project of Suzhou, the Key Project Supported by the Medical Science and Technology Development Foundation, Jiangsu Province Department of Health, the National and Local Engineering Laboratory of New Functional Polymer Materials, the Special Project of Diagnosis and Treatment for Clinical Diseases of Suzhou and the Jiangsu Medical Research Project.

## Additional information

### Funding

| Funder | Grant reference number | Author |
| --- | --- | --- |
| National Nature Science Foundation of China | 82072425 | Dechun Geng |
| National Nature Science Foundation of China | 82272567 | Dechun Geng |
| National Nature Science Foundation of China | 82072424 | Qianping Guo |
| Priority Academic Program Development of Jiangsu Higher Education Institutions | | Dechun Geng |
| Science and Technology Project of Suzhou | GSWS2020121 | Zhen Wang |
| Key Project Supported by the Medical Science and Technology Development Foundation | H2019024 | Minfeng Gan |
| Special Project of Diagnosis and Treatment for Clinical Diseases of Suzhou | LCZX202003 | Dechun Geng |
| National Nature Science Foundation of China | 82072498 | Yaozeng Xu |
| Natural Science Foundation of Jiangsu Province | BK2021650 | Yaozeng Xu |
| Science and Technology Project of Suzhou | GSWS2022002 | Dechun Geng |

| Funder | Grant reference number | Author |
| --- | --- | --- |
| National and Local Engineering Laboratory of New Functional Polymer Materials | SDGC2205 | Dechun Geng |
| Jiangsu Medical Research Project | ZD2022014 | Dechun Geng |

The funders had no role in study design, data collection and interpretation, or the decision to submit the work for publication.

## Author contributions

Gaoran Ge, Conceptualization, Data curation, Software, Visualization, Methodology, Writing – original draft; Qianping Guo, Resources, Investigation, Methodology; Ying Zhou, Investigation, Methodology; Wenming Li, Wei Zhang, Huaqiang Tao, Methodology; Jiaxiang Bai, Conceptualization, Investigation; Qing Wang, Wei Wang, Software, Methodology; Zhen Wang, Funding acquisition; Minfeng Gan, Resources; Yaozeng Xu, Huilin Yang, Resources, Funding acquisition; Bin Li, Conceptualization, Resources; Dechun Geng, Conceptualization, Resources, Funding acquisition, Project administration, Writing – review and editing

## Author ORCIDs

Gaoran Ge  https://orcid.org/0000-0002-5535-7483
Jiaxiang Bai  https://orcid.org/0000-0002-3485-5563
Dechun Geng  https://orcid.org/0000-0003-4375-2803

## Ethics

The samples were obtained with the informed consent of the patients, and all operations were approved by the Ethics Committee of the First Affiliated Hospital of Soochow University ((2018) Ethical Approval No.012).

All animal experiments were performed in strict accordance with the guide for the care and use of laboratory animals of Laboratory Animal Center of Soochow University and were approved by the Animal Ethics Committee of Soochow University (201910A354).

## Decision letter and Author response

Decision letter https://doi.org/10.7554/eLife.92142.sa1
Author response https://doi.org/10.7554/eLife.92142.sa2

# Additional files

## Supplementary files

- Supplementary file 1. siRNA target sequences.
- Supplementary file 2. Predictive analysis of protein binding.
- Supplementary file 3. RT-qPCR primer sequences.
- MDAR checklist

## Data availability

All data generated or analysed during this study are included in the manuscript and supplementary files. Source date files have been provided for all figures.

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
