## [Editor Report]

This study presents an important finding on the role of GLI1 in macrophages and osteoclasts in the pathogenesis of inflammatory arthritis, and suggests a therapeutic potential of GLI1 targeting in rheumatoid arthritis. The evidence supporting the claims of the authors is solid, although inclusion of the analysis of cell type-specific GLI1-deficient mice would have strengthened the study. The work will be of broad interest to immunologists, rheumatologists and bone biologists.

---

## [Decision Letter]

[Editors' note: this paper was reviewed by Review Commons.]

---

## [Author Response]

*General Statements [optional]*

We are very pleased to have been given the opportunity to revise our review manuscript. Thank you for your letter and the Reviewers’ comments on March 13, 2023 regarding our manuscript titled “GLI1 facilitates rheumatoid arthritis by collaborative regulation of DNA methyltransferases” (Manuscript #: RC-2023-01868). Those comments are all valuable and helpful for revising and improving our manuscript. Here, we have made a full revision on the manuscript to describe them more clearly and comprehensive.

We earnestly appreciate Editors/Reviewers’ warm work and hope that these revisions have improved the manuscript such that you and the Reviewers will deem it worthy of publication. Once again, thank you very much for your assistance. If there are any problems or questions about our manuscript, please do not hesitate to contact us at the address below.

*Point-by-point description of the revisions*

Reviewer #1 (Evidence, reproducibility and clarity (Required)):SummaryGe et al. defined the role of Gli1 in M1 macrophage activation and osteoclast differentiation in physiological conditions and inflammatory arthritis. The authors found that Gli1 expression is elevated in human RA synovial tissue relative to that in healthy donor controls. Moreover, the authors showed that the administration of GANT58, a Gli1 inhibitor, ameliorates inflammation and bone erosion in CIA mice. Gli1 expression is suppressed by LPS/IFN-γ stimulation in Raw264.7 cells while being induced by RANKL stimulation in Raw264.7 cells. However, GANT58 suppressed LPS/IFN-ɣ -induced expression of inflammatory cytokines and iNOS and osteoclastogenesis. The authors also identified DNMT1 and DNMT3a as downstream effectors of Gli1. Transcriptomic analysis of GANT58 treated Raw264.7 cells identified diminished protein expression of DNMT1 and DNMT3a by GANT58. Gli1 also directly interacts with DNMT1. Intriguingly, DNMT1 overexpression restores the effect of GANT58 on LPS/IFN-ɣ-mediated activation, while DNMT3a overexpression reverses the effect of GANT58 on RANKL-induced osteoclastogenesis. Since this study defines the role of Gli1 in the function and differentiation of myeloid cells, this is interesting. In addition, GANT58 nearly completely protects mice from arthritis, suggesting a therapeutic potential of Gli1 targeting in RA. However, the details of experiments are not clearly described, and the authors present the mixed data from Raw264.7 cells and BMMs without any explanations.

Many thanks for your recognition and constructive comments on our research. In this study, used mouse macrophage-like cell line RAW264.7 and primary bone marrow-derived macrophages (BMMs). The RAW264.7 is the most commonly used mouse macrophage cell line in medical research, and it is one of the most commonly used in vitro models for osteoclasts and inflammation research. In addition, compared with cell lines, primary cells have the characteristics of unchanged genetic material and biological characteristics closer to cell physiology in vivo. Therefore, in addition to cell lines, we also extracted primary macrophages from bone marrow for experiments to improve the reliability of this study. According to your comments, we have revised the manuscript, and our point-by-point responses are shown as follows.

Major commentsComment 1. Figures 1h and i. The author should show the histological score.

Thanks for the constructive comment. According to your suggestion, we have scored the results of H and E staining histologically and added quantitative results.

Comment 2. Pharmacological inhibitors often show non-specific effects. To complement their findings showing the effect of GANT58 on M1 macrophage activation and osteoclastogenesis, the authors should utilize Gli1-deficient cells that can be obtained by siRNAs-mediated knock down or Gli1 deletion.

Thanks for the professional and constructive comment. To make the results more reliable, we have synthesized siRNA and supplemented the related experiments to verify the role of GLI1 in M1 macrophage activation and osteoclastogenesis, which showed the same trend as GANT58 intervention. In the revised manuscript, the relevant results were shown in Figure S5:

Comment 3. Figure 4d: The authors should measure DNMT1 and DAMT3a RNA expression in LPS/IFN-ɣ- treated (Figure 2c and d) or RANKL treated Raw264.7 cells.

Thanks for your constructive comment. According to the suggestion, we have added the RNA expression of DNMT1 and DAMT3a to the revised Figure 4. At the same time, the corresponding contents are also described in the Results part.

Comment 4. The authors should provide detailed information of RNA-seq including how many genes are regulated by GANT58 and what is their cutoff (fold induction and FDR). The authors should deposit their RNA seq data in the public databases repository such as GEO.

Thanks for the professional and constructive comment. In the revised manuscript, we have made a more detailed analysis of the sequencing results and the detail information of RNA-seq have been added in the supplementary information.

Revised in the manuscript:

“2.4. GLI1 regulates the expression of DNMTs in distinct ways during the different fates of macrophages

As a nuclear transcription factor, GLI1 exerts an active effect through nuclear entry. In order to explore the potential downstream regulation mechanism of GLI1, RNA sequencing (RNA-seq) on the macrophages before and after GLI1 intervention was performed then to observe gene expression changes. The seq data showed that more genes were down-regulated (143) than up-regulated (74) in GANT58 treated cells (Figure S7a, b). Among these differentially altered genes, we revealed through Gene Ontology (GO) analysis that GANT58's intervention in GLI1 affected multiple biological processes including macrophage chemotaxis and macrophage cytokine production (Figure 4a). What’s more, the results of the Kyoto Encyclopedia of Genes and Genomes (KEGG) enrichment analysis of the pathway team enrichment was then performed and we showed the TOP30 enriched pathway. In these pathways, we classified them into cellular processes (red), human diseases (blue) and organismal systems (green) respectively. It showed that these down-regulated genes were involved in the development of human diseases such as rheumatoid arthritis, as well as organismal systems such as osteoclast differentiation (Figure S8c; Figure 4b). These evidences confirmed our previous results. Specifically, GANT58 reduced some of the osteoclast and inflammation-related genes in the cell resting state.”

Comment 5. Figure 5c. The authors should add non-stimulating condition as a control.

Thanks for your constructive comment. We have re-conducted the experiment and added the control group.

Comment 6. Figure 6C: DNMT3a deficiency regulates limited number of genes such as IRF8. The authors should measure IRF8 RNA or protein expression in RANKL-treated cells.

Thanks for your constructive comment. It is reported that DNMT3a can affect the activity of IRF8 and regulate the formation of osteoclasts. Thus, according to your suggestion, we have added IRF8 gene expression detection in the revised manuscript. As shown in Figure 6—figure supplement 1, the gene expression of *Irf8* was decreased after being treated by RANKL. However, the expression of Irf8 was reversed by Dnmt3a knock down.

Comment 7. Although the effects of Gli1 on bone metabolism in the literature are inconclusive, Gli1 is expressed on other cell types in bone. Gli1 haplodeficiency in mice decreased bone mass with reduced bone formation and enhanced bone resorption compared to control mice (PMID:25313900). Gli1 is also used as a marker for osteogenic progenitors which are precursors of chondrocytes and osteoblasts (PMID: 29230039). Thus, the beneficial effect of GANT58 on inflammation and bone erosion in CIA mice may result from the effects of GANT58 on multiple cell types other than F4/80+ cells. The authors should include these references in the discussion on pg.9 and expand their discussion.

Thank you for your constructive comments. Indeed. there have been some divergent conclusions about the function of hedgehog and GLI1 in bone metabolism, which suggests that GLI1 may have multiple roles. According to your suggestion, we have expanded the relevant discussion and added related references in the Discussion part.

Revised in the manuscript:

Discussion:

“Although we have demonstrated that the inhibition of GLI1 by GANT58 can reduce the inflammatory response and inhibit osteoclast formation and that this mechanism is achieved through the downregulation of DNMTs, these findings also raise new questions. In the previous research report, Gli1 haplodeficiency in mice decreased bone mass with reduced bone formation compared to control mice, which was due to the osteoblasts with weakened function [44]. In this process, the osteogenic differentiation of mesenchymal stem cells also affected the function of osteoclasts. In addition, GLI1 is also used as a marker for osteogenic progenitors which are precursors of chondrocytes and osteoblasts [45]. These studies suggest that the regulation of GLI1 on bone metabolism is complex, and the therapeutic effect of GANT58 on RA may be more than just affecting the inflammatory reaction mediated by macrophages and the bone destruction mediated by osteoclasts. In addition to macrophages and osteoclasts, the functions of synovial fibroblasts and osteoblasts play essential roles in the RA microenvironment. These cells are also closely linked to each other. Synovial fibroblasts OPG and RANKL secreted by osteoblasts are important factors that regulate osteoclasts. Therefore, in a follow-up study, we will extend the study of GLI1 to its regulatory mechanism in osteoblasts.”

Reference:

[44] Y. Kitaura, H. Hojo, Y. Komiyama, T. Takato, U.I. Chung, S. Ohba, Gli1 haploinsufficiency leads to decreased bone mass with an uncoupling of bone metabolism in adult mice, PLoS One 9(10) (2014) e109597.

[45] Y. Shi, G. He, W.C. Lee, J.A. McKenzie, M.J. Silva, F. Long, Gli1 identifies osteogenic progenitors for bone formation and fracture repair, Nat Commun 8(1) (2017) 2043.

Minor commentsComment 1. CIA model: The experiment design of CIA model is not clearly described. The author should specify the time point of GANT58 injection.

Thank you for your comment and we are sorry for the confusion caused by vague method descriptions about animal experiments. We have added the specific design and method description of related experiments in the revised manuscript.

Revised in the manuscript:

Materials and methods:

“An emulsion of bovine type II collagen (Chondrex, Redmond, WA, USA) and an equal amount (1:1, v/v) of complete Freund’s adjuvant (Chondrex) was prepared to establish the CIA mouse model. First, 0.1 ml of the emulsion was injected intradermally into the base of the tail on day 0. On day 21, 0.1 mg of bovine type II collagen mixed with incomplete Freund’s adjuvant (Chondrex) was injected. From the 21st day, mice began to receive injection intervention treatment. For vehicle group, mice were injected with the same volume of placebo daily. For treatment groups, mice were injected with GANT58 or 5-AzaC solution daily. All interventions began the day after the second injection of bovine type II collagen. Arthritis score was given every three days from the second immunization. On day 49, all mice were sacrificed (in accordance with the guidelines of the Animal Welfare and Ethics Committee of the Soochow University) for the collection of specimens.”

Comment 2. Joint inflammation of RA can be caused by many different cells. Abstract needs to be revised.

Thanks for your constructive comment. According to the suggestion, we have revised relevant descriptions in the abstract.

Revised in the manuscript:

Abstract:

“Rheumatoid arthritis (RA) is characterized by joint synovitis and bone destruction, the etiology of which remains to be explored. Many types of cells are involved in the progress of RA joint inflammation, among which the overactivation of M1 macrophages and osteoclasts has been thought an essential cause of joint inflammation and bone destruction. Glioma-associated oncogene homolog 1 (GLI1) has been revealed to be closely linked to bone metabolism. In this study, GLI1-expression in synovial tissue of RA patients showed to be positively correlated with RA-related scores and was highly expressed in collagen-induced arthritis (CIA) mouse articular macrophage-like cells. The decreased expression and inhibition of nuclear transfer of GLI1 downregulated macrophage M1 polarization and osteoclast activation, the effect of which was achieved by modulation of DNA methyltransferases (DNMTs) via transcriptional regulation and protein interaction ways. By pharmacological inhibition of GLI1, the proportion of proinflammatory macrophages and the number of osteoclasts were significantly reduced, and the joint inflammatory response and bone destruction in CIA mice were alleviated. This study clarified the mechanism of GLI1 in macrophage phenotypic changes and activation of osteoclasts, suggesting potential applications of GLI1 inhibitor in the clinical treatment of RA.”

Comment 3. Figure 4g, h: are these experiments done in the resting states?

Thank you for your comment. This part of the experiments was carried out during the induction of M1 macrophage or the induction of osteoclast. In this work, we found that GANT58 can inhibit GLI1 and at the same time reduce the gene expression of DNMT3a but not DNMT1 in the resting state. However, during M1 macrophage and osteoclast induction, GANT58 seemed to be able to inhibit both DNMT1 and DNMT3a protein expression. In view of the discovery that the expression of DNMT1 increased during the polarization of M1 macrophages, while the expression of DNMT3a increased during the activation of osteoclasts, we performed the binding experiment of GLI1 with DNMT1 in the process of LPS/IFN-γ induction, while the binding experiment with DNMT3a in the process of RANKL induction. We have added a detailed description to the revised manuscript.

Reviewer #1 (Significance (Required)):Strengths: Hedgehog (hh) signaling has been implicated in the differentiation of osteogenic progenitors. Gli1+ mesenchymal progenitors are responsible for both normal bone formation and fracture repair. This study defines a new role of Gli1 in the function and differentiation of myeloid cells. In addition, GANT58 nearly completely protects mice from arthritis, suggesting a therapeutic potential of Gli1 targeting in RA.

Thank the reviewer for your recognition of our research work.

Limitations: This study mainly uses a pharmacological inhibitor to study the mechanism underlying Gli1's action. In addition, the details of experiments are not clearly described, and the authors present the mixed data from Raw264.7 cells and BMMs without any explanations. Advance: This study provides conceptual advancement for hh signaling research by demonstrating the function of Gli1 in myeloid cells.

Thank the reviewer for your constructive comments and help us to further improve the manuscript.

Audience: Basic researchReviewer #2 (Evidence, reproducibility and clarity (Required)):Summary:The paper by Ge et al. seeks to identify a role for GLI1 in rheumatoid arthritis, as GLI1 is upregulated in the synovium of patients with rheumatoid arthritis. Inhibition of GLI1 by the GANT58 limited inflammation and destructive bone loss in a murine model of arthritis (Collagen Induced Arthritis). Inhibition of GLI1 increased expression of pro-inflammatory cytokines and M1 macrophage differentiation. Inhibition of GLI1 also blocked osteoclast formation. As has been shown in other settings, the function of GLI1 in M1 and osteoclast differentiation was linked to regulation by DNMTs.Major comments:Comment 1. There are several main problems with the text. Overall, the authors show an intriguing set of data implicating the use of GANT58 as a means to limit rheumatoid arthritis inflammation and bone destruction. The authors directly link the functions of GANT58 with loss of GLI1 activity by showing that GLI1 protein is reduced or translation to the nucleus blocked. It would be compelling if the authors would leverage a genetic model (either GLI1 knockout, or a CRISPR/siRNA approach) to see if it recapitulates key findings in vitro and in vivo. These data could further their claims that their findings are in fact directly due to GLI1.

Thanks for the professional and constructive comment. To make the results more reliable, we have synthesized siRNA and supplemented the related experiments to verify the role of GLI1 in M1 macrophage activation and osteoclastogenesis. Related experiments have been updated in the revised manuscript.

Comment 2. Overall, the paper lacks methodologic clarity that limits thorough interpretation of the data. Multiple experiments are missing from the Materials and methods, including descriptions of the definition of trabecular bone and its analysis in micro-CT, the means by which cytoplasmic and nuclear fractions were generated, and the timing and dosing of GANT58 in vitro studies. In addition, key details regarding the reagents include the sources of primary antibodies used in the western blots and immunoprecipitation studies. Important methodologies are not well explained, which include the treatment of the Sham animals (presumably healthy) are not explained, that is, whether they receive injections of vehicle or are truly naïve. Finally, there is no statistical methodology, minimal explanation of the RNA-sequencing analyses, and no statement about how the RNA-sequencing data will be made available. This lack of detail makes a thorough assessment of the quality and interpretations of the data challenging and replication of the results impossible.

Thanks for your careful reading and constructive comments. We are sorry for the lack of some detailed methodological descriptions in the manuscript. In order to better explain how our experiment is carried out and improve the repeatability of the experiment, we have comprehensively improved the description of the experimental method in the revised manuscript.

Revised in the manuscript:

Materials and methods:

“4.1. Experimental animals and human synovial tissue. Male DBA mice aged 6-8 weeks and weighing 15-20 g were randomly selected and fed in a specific pathogen-free (SPF) environment at a room temperature of 25℃, a relative humidity of 60%, and 12 hours of alternating light. All animal experiments were approved by the Animal Ethics Committee of the Soochow University (201910A354). The animals were divided randomly into groups (6 per group): sham group (healthy mice not received any treatment), vehicle control group (CIA model mice treated with solvent), and GANT58 (GLI1 specific inhibitor; MedChemExpress, New Jersey, USA) group (mice treated with 20 mg/kg GANT58) or 5-AzaC (DNMTs specific inhibitor; MedChemExpress) group (mice treated with 2 mg/kg 5-AzaC). An emulsion of bovine type II collagen (Chondrex, Redmond, WA, USA) and an equal amount (1:1, v/v) of complete Freund’s adjuvant (Chondrex) was prepared to establish the CIA mouse model. First, 0.1 ml of the emulsion was injected intradermally into the base of the tail on day 0. On day 21, 0.1 mg of bovine type II collagen mixed with incomplete Freund’s adjuvant (Chondrex) was injected. For vehicle group, mice were injected with the same volume of placebo daily. For treatment groups, mice were injected with GANT58 or 5-AzaC solution daily. All interventions began the day after the second injection of bovine type II collagen. Arthritis score was given every three days from the second immunization. On day 49, all mice were sacrificed (in accordance with the guidelines of the Animal Welfare and Ethics Committee of the Soochow University) for the collection of specimens.”

“4.3. Micro-CT analysis. The fixed bone samples of mice were collected. The joint samples were placed in a SkyScan 1174 Micro-CT scanning warehouse (Belgium). The parameters were set as follows: voltage 50 kV, current 800 μA, scanning range 2 cm × 2 cm, and scanning layer thickness 8 μm. The scan data were then entered into computer to conduct three-dimensional reconstruction with NRecon software (Bruker, Germany), and the bone tissue parameters were analysed with CTAn software (Bruker, Germany) after data conversion. During this procedure, we performed an analysis of bone parameters including BMD (Bone Mineral Density), BV/TV (Percentage Trabecular Area), Tb.N (Trabecular Number) and Tb.Sp (Trabecular Separation) by selecting the small joint of paws as the region of interest (ROI) in CTAn software. The three-dimensional reconstruction images were exhibited by Mimics Research software (Version 21.0; Materialise, Belgium).”

“4.11. Western blotting. Cells were seeded in 6-well plates at a density of 1 × 10^6^/well with stimulation with RANKL (50 ng/ml) or LPS (100 ng/ml) + IFN-γ (20 ng/ml). First, cells were collected to extract total protein, and the BCA (Beyotime) method was used to adjust the protein concentration. Total protein was mixed with 5× loading buffer (Beyotime) and boiled at 95 °C for 10 minutes. For cytoplasmic/nucleus isolation, cells were collected and protein was extracted according to the instructions using the nuclear protein and cytoplasmic protein extraction kit (Beyotime). The proteins were separated by SDS polyacrylamide gel electrophoresis (SDS–PAGE; EpiZyme, Shanghai, China) based on their different molecular weights. Electrophoresis was performed using Bio–Rad (California, USA) equipment at 180 V for 40 minutes. Then, the proteins were transferred to a nitrocellulose membrane at 350 mA for 70 minutes using membrane transfer equipment (Bio–Rad). The membrane was removed and placed into western blot blocking buffer for 1 hour at room temperature. The diluted primary antibodies (GLI1, Abclonal, A14675; β-actin, Beyotime, AF5003; Lamin-B1, Abcam, ab16048; NFATc1, Abclonal, A1539; CTSK, Abclonal, A5871; MMP9, Abclonal, A11147; DNMT1, Abclonal, A16729; DNMT3a, Cell Signaling Technology, D23G1; GAPDH, Abclonal, A19056) were placed on the membrane and incubated at 4 ℃ for 12 hours, and then the corresponding secondary antibody was added and incubated for 1 hour at room temperature. Finally, a chemiluminescence detection system (Bio–Rad) was used to observe the results.”

“4.12. High-throughput sequencing (RNA-seq). To further screen for differential genes, we first subjected RAW264.7 cells to a 24-hour adaptive culture, followed by the addition of GANT58 at a final concentration of 10 μM to the GANT58 intervention group and cultured for a total of 24 h. After the cell treatment was completed, cells of the control group and GANT58 treated group were collected respectively, and RNA-seq detection and analysis were entrusted to a professional biological company (Azenta Life Sciences, Suzhou, China). Briefly, for differential expression gene analysis, the differential expression conditions were set as fold change (FC) > 1.5 and false discovery rate (FDR) < 0.05. Among the differential genes, we performed GO analysis of the biological process and showed the TOP20 enriched biological activities. KEGG analysis of the pathway team enrichment was then performed and we showed the TOP30 enriched pathway. In these pathways, we classified them into Cellular Processes (red), Human Diseases (blue) and Organismal Systems (green) respectively and showed the enrichment of TOP5 in each category.”

“4.14. Statistical analysis. All data are presented as the mean ± standard deviation (SD). Statistical analysis was performed with an unpaired two-tailed Student’s t test for single comparisons with GraphPad Prism 8 (GraphPad Software, CA, USA). One-way analysis of variance (ANOVA) was used to compare data from more than two groups. p values less than 0.05 were considered statistically significant.”

The specific statistical methods are marked in Figure legends as well.

Data Availability: The authors declare that all data supporting the findings of this study are available within this paper and its Supplementary Information and raw data are available on request from the corresponding author.

Comment 3. The authors should expand their introduction and Discussion to include a description of the history of other GLI inhibitors (such as GANT61) in rheumatoid arthritis. Further, the authors failed to cite current studies showing that GLI1 is upregulated in RA patients (DOI: 10.1007/s10753-015-0273-3 amongst others).

Many thanks to your thoughtful reading and constructive comment. According to your suggestion, we have added some revisions, including the description of GLI1 inhibitors, in the introduction and Discussion sections. At the same time, we have also added descriptions and citations of GLI1 and RA-related research in corresponding positions.

Revised in the manuscript:

Introduction:

“To date, three mammalian GLI proteins have been identified, among which GLI1 usually acts as a transcriptional activator. On the basis of these studies, small molecular compounds such as GANT58 (selective inhibitor of GLI1) and GANT61 (inhibitor of GLI1 and *GLI2*) are often used as pharmacological interventions of GLI1, so as to achieve the purpose of inhibiting GLI1 activity and regulating the molecular biological process [13, 14]. Many of the physiopathological processes involved with GLIs are complex and worth discussing. Relevant studies have shown that GLI1-activated transcription promotes the development of inflammatory diseases such as gastritis, and antagonizing GLI1 transcription can alleviate the inflammatory degradation of articular cartilage [15, 16].”

Discussion:

“In previous studies, GLI1 signal transduction and other pathways, including the NF-κB signaling pathway, were usually studied in tumor-associated diseases and are considered a response network that promotes cancer development [21, 22]. Qin. et al. found that the content of SHH in RA patients serum increased significantly by comparing with healthy patients [23]. At the same time, our study also showed that GLI1 was more expressed in the joint tissue of RA patients. These results suggest that HH-GLI signaling pathway may be involved in the regulation of the pathological process of RA. However, the research results of the hedgehog pathway in bone metabolism are complex.”

Reference:

[13] X. Chen, C. Shi, H. Cao, L. Chen, J. Hou, Z. Xiang, K. Hu, X. Han, The hedgehog and Wnt/β-catenin system machinery mediate myofibroblast differentiation of LR-MSCs in pulmonary fibrogenesis, Cell Death Dis 9(6) (2018) 639.

[14] R.K. Schneider, A. Mullally, A. Dugourd, F. Peisker, R. Hoogenboezem, P.M.H. Van Strien, E.M. Bindels, D. Heckl, G. Busche, D. Fleck, G. Muller-Newen, J. Wongboonsin, M. Ventura Ferreira, V.G. Puelles, J. Saez-Rodriguez, B.L. Ebert, B.D. Humphreys, R. Kramann, Gli1(+) Mesenchymal Stromal Cells Are a Key Driver of Bone Marrow Fibrosis and an Important Cellular Therapeutic Target, Cell Stem Cell 23(2) (2018) 308-309.

[23] S. Qin, D. Sun, H. Li, X. Li, W. Pan, C. Yan, R. Tang, X. Liu, The Effect of SHH-Gli Signaling Pathway on the Synovial Fibroblast Proliferation in Rheumatoid Arthritis, Inflammation 39(2) (2016) 503-12.

Comment 4. The antibody for GLI1 seems poor and inconsistent. Knockdown studies to show its specificity, and an example of the whole membrane stained for GLI1 would provide important validation of the reagent.

Thanks for your comment and we are sorry for showing the western blot results with poor quality. In the revised manuscript, we used the newly purchased antibody (Abclonal, Catalog: A14675) and rearranged the groupings for better comparison of protein expression and replaced the results with clearer blot images. Original images of all western blot results can be uploaded subsequently.

Comment 5. Regarding Figure S1:The studies of RA patients are underpowered. With only three RA patients and three healthy synovial the distribution of DAS28 scores is clustered at healthy and active disease, and the correlation study is unconvincing.

Thanks for your constructive comment. We are sorry that the studies of RA patients might not be convincing enough due to the small sample size. In order to avoid controversial conclusions, we left out the results of correlation analysis between GLI1 expression and DAS28. In the follow-up study, we will collect additional clinical pathology data for statistical analysis and quantified the expression of GLI1 in healthy control patients and RA patients.

Comment 6. Regarding Figure 1 f-g and Figure 4j-k:However, the information on inflammatory bone loss are incomplete. The methodology for the assessment of BMD and trabecular bone parameters in the hind paw is not explained. The 3D reconstructions are of the whole bone hind paw, but the anatomical region where trabecular bone is assayed not defined. It would be convincing if the authors added erosion scores in the hind paws or knees to show that the erosion in the synovium, which contributes to inflammatory arthritis, mirrors what occurs in the trabeculae.

Thanks for your constructive comment. We are sorry for incomplete description on in vivo experiments, including the micro-CT analysis and histological analysis. In the revised manuscript, we further supplemented and improved the relevant methods. The Inflammatory cell infiltration score and bone erosion score were also added according to your suggestion.

Revised in the manuscript:

Materials and methods:

“4.3. Micro-CT analysis. The fixed bone samples of mice were collected. The joint samples were placed in a SkyScan 1174 Micro-CT scanning warehouse (Belgium). The parameters were set as follows: voltage 50 kV, current 800 μA, scanning range 2 cm × 2 cm, and scanning layer thickness 8 μm. The scan data were then entered into computer to conduct three-dimensional reconstruction with NRecon software (Bruker, Germany), and the bone tissue parameters were analysed with CTAn software (Bruker, Germany) after data conversion. During this procedure, we performed an analysis of bone parameters including BMD (Bone Mineral Density), BV/TV (Percentage Trabecular Area), Tb.N (Trabecular Number) and Tb.Sp (Trabecular Separation) by selecting the small joint of paws as the region of interest (ROI, bone tissue from ankle joint to toe) in CTAn software. The three-dimensional reconstruction images were exhibited by Mimics Research software (Version 21.0; Materialise, Belgium).”

Comment 7. Regarding Figure 2:– The methods and text do not state the dose of GANT58 used in these assays. Nor do they specify the timing of the GANT58 application in relationship to LPS and IFNγ stimulation.

Thanks for your thoughtful reading and constructive comment. We apologize for not expressing the detailed dose and intervention time of GANT58 in some experiments in detail. In the revised manuscript, we have added drug dose and intervention time cutoff points in the parts of Methods, Results, and Figure Legends.

– The authors conclude that GLI1 limits the differentiation of M1 macrophages and also directly blocks the production of pro-inflammatory cytokines. The data are difficult to parse in that the directionality is not clear. If GLI1 promotes M1 macrophages, there would be less proinflammatory cytokines due to the reduction of their proliferation. To evaluate the role of GLI1 in regulating the cytokines, additional studies showing a transcriptional regulation of these cytokines is warranted.

Thank you for your professional and constructive comment. We totally agree with you that the release of inflammatory cytokines is affected not only by gene expression but also by the number of cells that proliferate. Therefore, to exclude this interference, we further examined transcriptional expression of cytokines responsible for cellular inflammation under the same conditions. The results shown in Figure 2—figure supplement 3 confirmed the inhibition of GANT58 on the expression of pro-inflammatory cytokine mRNAs, which further supported our conclusion.

– To show that the fractionation of the cytoplasm and nuclear compartments was complete, the westerns for GLI1, lamin-B1 and β actin should be shown in the same blot.

Thank you for your professional and constructive comment. According to your suggestion, we have rearranged the groupings to show the westerns for GLI1, lamin-B1 and β-actin in the same blot for better comparison.

– In Section 2.3 ("the expression of and intranuclear transport…"), the authors state that their previous studies showed GLI was expressed in macrophages (line 80-81). It is unclear whether the authors are referring to studies in this manuscript or a previously published study and a citation is needed.

Thank you for your careful reading and helpful comment. We are sorry that the description in this part is confusing. In fact, what we want to refer to is the in vivo results described in the first section of the results part. We have changed this description in the revised manuscript.

Revised in the manuscript:

“2.3. The expression and intranuclear transport of GLI1 is involved in osteoclast activation

The over activation of osteoclast is the direct cause of bone destruction in RA. As described of the in vivo experimental results in the first part, we have found that GLI1 is highly expressed in macrophage-like cells in the subchondral bone of the joints, which raised our concerns about GLI1 and osteoclasts.”

In response to Figure 3:-The authors show that GANT58 has a potent impact in limiting osteoclast formation. The text states that GANT58 is a pretreatment, but the timing of this is not stated.

Thanks for your constructive comment. In order to reach the working concentration of drugs at the beginning of some experiments, we usually pretreated cells for 6-8 hours. We have added the specific time in the parts of Materials and methods or Figure legends.

– It would be interesting to see whether there is a dose-response effect of GANT58.

Thanks for your comment. According to your comment, we set the concentration of GANT58 to 0, 1, 5 and 10 μM to intervene the induction of M1 macrophages and osteoclasts respectively. As shown in Author response image 1, with the increase of GANT58 concentration, the mean fluorescence intensity of iNOS in macrophages seems to decrease gradually, but there is no statistical significance when the concentration is below 5 μM. Similarly, when the concentration reached 10μM, GANT58 significantly inhibited the formation of osteoclasts.

**Author response image 1. sa2fig1:** (a) M1 macrophages were induced for 24 h with intervention of different concentrations of GANT58. Immunofluorescence staining of iNOS. (b) Osteoclast were induced for 4 days with intervention of different concentrations of GANT58. TRAP staining images and TRAP positive osteoclast number quantification results. Data shown represent the mean ± SD. *p < 0.05, ns = no significance.

– It is not stated how long the cells are RANKL treated prior to nuclear/cytoplasmic fractionation? (3a, b, c and i).

Thanks for your constructive comment. For osteoclast induction and intervention, we treated cells for 48 h as cell transcription regulation usually occurs in the early and middle stages of osteoclast differentiation. According to your comment, we have added the description of specific intervention time information in Figure legends and other parts.

– The "Zoom" images in Figure 3j do not have a box to delineate where the higher magnification images are taken from in the top panes. The images appear to be from serial sections. This should be clarified.

Thanks for your constructive comment. In the revised Figure, we have boxed the area represented by the Zoom images. We can ensure that these images come from different groups of specimen slices. In order to better observe the number of osteoclasts, we chose a larger shooting multiple, which might make the pictures look similar. The revised images are shown in the Figure 3n, o.

In Figure 3 and Figure 6e and 6f:Although the data in BMM showed that there was no impact on cell survival was limited at low concentrations, showing that the differentiating osteoclasts are not more sensitive to apoptosis by GANT58 would be compelling. The large difference in cellularity in the presence of GANT58 provokes this question.

Thank you for your careful reading and helpful comment. As shown of the CCK8 result, GANT58 had no significant inhibitory effect neither on BMMs nor RAW264.7 cells until the concentration reached 40 μM. In the process of changing the polarization phenotype of macrophages, the cell morphology will also change to some extent. In our research results, the change of cell morphology after GANT58 intervention might be due to the inhibition of M1 macrophages. In order to observe the effect of GANT58 on BMM cell death and apoptosis, we further performed living/dead staining and apoptosis detection by fluorescence after GANT58 intervention. The results showed that GANT58 did not change the level of apoptosis nor increase the number of dead cells at the concentration of 10 μM. However, when the concentration increased to 30μM, the number of apoptotic cells increased. These results suggest that we should pay strict attention to the control of drug concentration in experimental intervention and transformation application. The supplementary results are shown in .

In Figure 4:– The IP studies (4g and 4h) lack showing successful pull-down of GLI1 by western blotting as a critical control for the study.

Thanks for your constructive comment. In fact, during the performance of CO-IP experiment, we simultaneously detected the expression of GLI1 to verify the effectiveness of the antibodies used. According to your comment, In the revised Figure 4g and h, we have updated these corresponding results.

**–** Details about the steps involved in RNA-sequencing analyses need to be provided.

Thanks for your constructive comment. According to your suggestion, we have provided the steps involved in RNA-sequencing analyses in the Methods.

“4.12. High-throughput sequencing (RNA-seq). To further screen for differential genes, we first subjected RAW264.7 cells to a 24-hour adaptive culture, followed by the addition of GANT58 at a final concentration of 10 μM to the GANT58 intervention group and cultured for a total of 24 h. After the cell treatment was completed, cells of the control group and GANT58 treated group were collected respectively, and RNA-seq detection and analysis were entrusted to a professional biological company (Azenta Life Sciences, Suzhou, China). Briefly, for differential expression gene analysis, the differential expression conditions were set as fold change (FC) > 1.5 and false discovery rate (FDR) < 0.05. Among the differential genes, we performed GO analysis of the biological process and showed the TOP20 enriched biological activities. KEGG analysis of the pathway team enrichment was then performed and we showed the TOP30 enriched pathway. In these pathways, we classified them into Cellular Processes (red), Human Diseases (blue) and Organismal Systems (green) respectively and showed the enrichment of TOP5 in each category.”

– Studies have previously shown a reduction of inflammatory arthritis by 5'-Azac and should be cited.

Thank you for your careful reading and helpful comment. In the discussion part of the revised manuscript, we have cited the related articles, which is shown as below.

Revised in the manuscript:

Discussion:

“In addition to normal physiological development, the abnormal expression of DNMTs causes the development of tumors and other diseases [35]. Through the treatment of DNMTs inhibitors, the inflammatory arthritis in mice was significantly relieved, which was consistent with the previous studies [36]. These results suggested that DNMTs might be involved in the inflammatory reaction and bone destruction of RA. Reports have suggested that the absence of DNMT3a inhibits the formation of osteoclasts, which may be due to the methylation of downstream IRF8 by DNMT3a [37]. In our study, we also verified this finding through pharmacological and genetic intervention.”

Reference:

[36] D.M. Toth, T. Ocsko, A. Balog, A. Markovics, K. Mikecz, L. Kovacs, M. Jolly, A.A. Bukiej, A.D. Ruthberg, A. Vida, J.A. Block, T.T. Glant, T.A. Rauch, Amelioration of Autoimmune Arthritis in Mice Treated With the DNA Methyltransferase Inhibitor 5'-Azacytidine, Arthritis Rheumatol 71(8) (2019) 1265-1275.

– What is the proposed functional consequence for GLI1 binding to DNMT3a? Does GLI1 inhibition lead to hypomethylation of DNA by DNMT?

Many thanks for your constructive comment. In this study, it is interesting to find that GLI1 can affect the expression of *Dnmt3a* at the level of gene transcription, and affect the expression of DNMT3a and DNMT1 both in the process of protein expression. Through the CO-IP experiment, we confirmed that GLI1 protein can bind to DNMT1 instead of DNMT3a protein. These results suggested that GLI1 may regulate the expression of DNMT3a and DNMT1 at genetic level and post-translation proteinic level, respectively. Patricia Gonz á lez Rodr í Guez's latest research showed that during autophagy induction, GLI1 is upregulated, phosphorylated, translocated to the nucleus and recruited to the regions closer to the Transcription Start Site (TSS) of the *Dnmt3a* gene. This may be the direct mechanism of GLI1 regulating the expression of DNMT3a [1]. Theoretically, the expression of DNMTs affects the degree of methylation of related genes [2]. Thus, in the follow-up study, we will further verify the degree of genomic methylation caused by GLI1's regulation of DNMTs, and further explore more possible ways of GLI1's regulation of DNMTs and its potential role in other cell models.

Reference:

[1] P. Gonzalez-Rodriguez, M. Cheray, L. Keane, P. Engskog-Vlachos, B. Joseph, ULK3-dependent activation of GLI1 promotes DNMT3A expression upon autophagy induction, Autophagy (2022) 1-12.

[2] Dura M, Teissandier A, Armand M, Barau J, Lapoujade C, Fouchet P, Bonneville L, Schulz M, Weber M, Baudrin LG, Lameiras S, Bourc'his D. DNMT3A-dependent DNA methylation is required for spermatogonial stem cells to commit to spermatogenesis, Nat Genet 54(4) (2022) 469-480.

Figure 5:The groups in 5g are not well defined.

Thank you for your careful reading and comment. We're sorry that we didn’t clearly show the grouping information. In the revised Figure 5g, we have added the complete information of the groups.

– DNMT1 and DNMT3a reduction by siRNA, CRISPR or knockout would strengthen the inhibitor studies.

Thanks for your constructive comment. In the revised manuscript, we knocked down the expression of DNMT1 and DNMT3a by siRNA, and supplemented the related experimental results, which are shown in Figure 5—figure supplement 2 and Figure 5—figure supplement 3.

Regarding Figure 5 and 6:-What is the impact of DNMT1 and DNMT3a overexpression on their own (not in the presence of GANT58)?

Thanks for your constructive comment. According to your comment, we observed and compared the differences in the polarization of macrophages M1 and the activation of osteoclasts between the DNMTs overexpression group and the control group. The results showed that overexpression of DNMT1 seemed to have no obvious effect on the formation of M1 macrophages. During the osteoclast activation, at day 4 of RANKL induction, the TRAP positive stained osteoclast number seemed to be no significance between WT group and *Dnmt3a^OE^* group. However, at day 3, there was more osteoclast in *Dnmt3a^OE^* group, which suggested that overexpression of *Dnmt3a* might accelerate the activation of osteoclasts to some extent. The results are shown in Figure 5—figure supplement 4 and Figure 6—figure supplement 2.

Minor comments:Comment 1. The authors do not include a description of DNMTs in the introduction.

Thanks for your constructive comment. According to your suggestion, we have added a description of DNMTs in the Introduction.

Introduction:

“DNA methylation is an important epigenetic marker playing an important role in regulating gene expression, maintaining chromatin structure, gene imprinting, X chromosome inactivation and embryo development an important epigenetic modification way to regulate gene expression, which is activated by DNA methyltransferases (DNMTs) [17]. As reported, DNMT1 and DNMT3a are involved in the progress of many physiological disorders, such as immune response and cell differentiation [18, 19]. In this study, …”

Reference:

[17] E. Li, Y. Zhang, DNA methylation in mammals, Cold Spring Harb Perspect Biol 6(5) (2014) a019133.

[18] Y. Fu, X. Zhang, X. Liu, P. Wang, W. Chu, W. Zhao, Y. Wang, G. Zhou, Y. Yu, H. Zhang, The DNMT1-PAS1-PH20 axis drives breast cancer growth and metastasis, Signal Transduct Target Ther 7(1) (2022) 81.

[19] R. Ramabadran, J.H. Wang, J.M. Reyes, A.G. Guzman, S. Gupta, C. Rosas, L. Brunetti, M.C. Gundry, A. Tovy, H. Long, T. Gu, S.M. Cullen, S. Tyagi, D. Rux, J.J. Kim, S.M. Kornblau, M. Kyba, F. Stossi, R.E. Rau, K. Takahashi, T.F. Westbrook, M.A. Goodell, DNMT3A-coordinated splicing governs the stem state switch towards differentiation in embryonic and haematopoietic stem cells, Nat Cell Biol 25(4) (2023) 528-539.

Comment 2. The descriptions of the groups are often unclear. In Figure 2, the label "GANT58" (blue bars) is presumably for a group that is treated for LPS+IFNγ+GANT58 but this is not clarified.

Thanks for your careful reading and we are sorry for the ambiguous labeling. We have checked the whole manuscript and changed the related labeling information.

Comment 3. The distinction of Figure 3g as multinuclear giant cells (vs TRAP+ OCs in panel 3d) should be explained.

Thanks for your comment. Osteoclast is defined as a multinucleated giant cell with bone absorption function, which is composed of multiple monocytes/macrophages [1]. As osteoclasts mature, their cytoskeleton will undergo drastic reorganization. Filamentous actin (F-actin) firstly constitutes a podosomes with a highly dynamic structure, thereby completing the cell adhesion, migration, dissolution of bone minerals and digestion of organic matrix [2]. Therefore, in addition to observing the formation of osteoclasts by TRAP staining, we also carried out immunofluorescence staining to observe the F-actin ring formation to further evaluate the functional maturity of osteoclasts. Osteoclasts usually have 2-50 nuclei, so we mainly regarded multinucleated giant cells with complete F-actin rings as mature osteoclasts during the quantification process.

Reference:

[1] da Costa CE, Annels NE, Faaij CM, Forsyth RG, Hogendoorn PC, Egeler RM, Presence of osteoclast-like multinucleated giant cells in the bone and nonostotic lesions of Langerhans cell histiocytosis. J Exp Med 7;201(5) (2005) 687-93.

[2] Portes M, Mangeat T, Escallier N, Dufrancais O, Raynaud-Messina B, Thibault C, Maridonneau-Parini I, Vérollet C, Poincloux R, Nanoscale architecture and coordination of actin cores within the sealing zone of human osteoclasts, *ELife* (11) (2022) e75610.

Comment 4. The labels in 4C of "R1, R2, R3" standing for GANT58 is confusing

We are sorry for the confusing labeling. In the revised manuscript, we have added specific grouping information in the Figure legend, as shown below.

“Figure 4. DNA methyltransferases might be a regulatory target downstream of GLI1. a Biological process GO analysis of RNA-seq results for macrophages with or without GANT58 treatment. b KEGG rich analysis of RNA-seq results. c Heat map of parts of the relevant gene transcriptional expressions (C = control group; R = GANT58 treated group; red: increased expression; blue: decreased expression). d Relative mRNA expression of Gli1, Dnmt1 and Dnmt3a in macrophages with or without GANT58 treatment. Statistical analysis was performed using two-way ANOVA test. e RAW264.7 cells were stimulated by LPS and IFN-γ for 24 h, with or without GANT58 co-intervention. Western blot results of DNMT1 and DNMT3a protein expression and grayscale value ratio to β-actin of western blot results. n=3. f RAW264.7 cells were stimulated by RANKL for 3 days, with or without GANT58 co-intervention. Western blot results of DNMT1 and DNMT3a protein expression and grayscale value ratio to β-actin of western blot results. n=3. Statistical analysis was performed using two-way ANOVA test. g, h Co-IP detection of protein binding between GLI1 and DNMT1/DNMT3a. n=3. i Protein–protein interface interaction of GLI1 and DNMT1 with PyMOL. j Micro-CT scanning and 3D reconstruction of mouse paws. k Bone parameters of BV/TV, BMD, Tb.N, Tb.Th. n=6. Statistical analysis was performed using one-way ANOVA test. Data shown represent the mean ± SD. *p < 0.05, **p < 0.01, ns = no significance.”

Comment 5. In Figure S8, the numbers between the western blots are not explained.

Many thanks for your careful reading and comment. The numbers between the blots represent the ratio of the gray value of DNMT1 and DNMT3a immunoblot to the gray value of β-actin immunoblot, so as to reflect the relative expression of proteins. In order to avoid confusion, we made a statistical chart of the results and added it to revised Figure S8.

Comment 6. In Figure S9 there are references to asterisks which do not appear in the figure.

We are sorry for the mistake. We have deleted the relevant information in the revised Supplementary information. Thanks again.

Reviewer #2 (Significance (Required)):The paper presented by Ge et al. present interesting data suggesting that a GLI1 inhibitor (GANT58) has a strong impact on inflammatory arthritis in a murine model. Interesting data are presented whose novelty need better contextualization with other published studies, as previously published studies which are not cited in this manuscript include the finding that GLI1 is upregulated in patients with rheumatoid arthritis, that other GLI inhibitors have been utilized in murine models of rheumatoid arthritis, and that GLI1 has been shown to regulate DNMT expression in cancer settings. The authors connect GLI1 inhibition with DNMT activation in limiting M1 macrophage and osteoclast differentiation. However, several important controls are needed to in the in vitro studies as outlined above.Reviewer #3 (Evidence, reproducibility and clarity (Required)):SummaryThe manuscript by Ge et al. describes the possible roles of GLI1 in macrophage and osteoclast activation in rheumatoid arthritis via its functional interaction with DNA methyltransferases. The authors found that the GLI1 expression was elevated in RA synovial tissues and GLI1-specific inhibitor, GANT58, ameliorated arthritis in CIA mice. GLI1 expression in F4/80-positive macrophages in CIA synovial tissues led the authors to assess the roles of GLI1 in macrophages and osteoclasts. GANT58 suppressed M1 macrophage polarization by IFNγ+LPS and osteoclastogenesis by RANKL. RNA-seq analysis of GANT58-treated macrophages revealed that DNA methyltransferases, DNMT1 and DNMT3a were possible targets of GLI1, and the studies with small inhibitors or overexpression of DNMTs suggest that GLI1 enhanced M1 polarization and osteoclastogenesis through DNMTs. The manuscript is well-written, the methods are accurate, and the results and data interpretation are consistent and clearly presented. This work deserves publication in Research Commons after addressing the following questions:Major commentsComment 1. GANT58 may inhibit GLI2 in addition to GLI1 and have off-target effects. Major findings with GANT58 in vitro, the suppressive effects on M1 polarization, osteoclastogenesis, and DNMT3a expression should be assessed with siRNA/shRNA knockdown or CRISPR/Cas9 knockout of GLI1.

Many thanks for your careful reading and constructive comment. According to your comment, we have constructed *Gli1* knock-down cells and carried out related experiments. The results have been added in the revised manuscript, which are shown in Figure 2L, Figure 3J/K, Figure 2—figure supplement 2 and Figure 4—figure supplement 2.

Comment 2. In CIA with GANT58, the author performed only preventive treatment, not therapeutic treatment. Does GANT58 suppress adaptive immune responses via inhibiting APC function (ex. anti-CII IgG production)? Alternatively, the inhibitory effects of GANT58 on the effecter phase of RA (M1 macrophage and osteoclast activation) can be assessed using the serum-transfer arthritis models.

Many thanks for your constructive comments. Your question is indeed a direction worthy of attention. In our study, GANT58 was given during the stage of model establishment, showing a good effect of relieving arthritis, which was proved to come from the direct inhibition of inflammatory phenotype macrophages and osteoclasts. However, as autoimmune diseases, the enhancement of antigen presenting function and anti-Col II IgG production can enhance the immune response of the body [1]. The regulatory effect of GANT58 on macrophages suggests that it may have a potential impact on APC function. Despite this, whether GANT58 can regulate the pathological process of RA by influencing this pathway is inconclusive. Therefore, according to your suggestion, we will improve the relevant experiments in the follow-up research, and apply GANT58 to various animal models of RA to further explore the possible mechanism of GANT58 in the treatment of RA and provide more reliable theoretical support for its transformation and application.

Reference:

[1] Tsark EC, Wang W, Teng YC, Arkfeld D, Dodge GR, Kovats S, Differential MHC class II-mediated presentation of rheumatoid arthritis autoantigens by human dendritic cells and macrophages, J Immunol 1;169(11) (2002) 6625-33.

Minor commentsComment 1. GANT58 is a water insoluble agent. Can you please include how to dissolve GANT58, administration route, and rationale of 20 mg/kg, for CIA?

Thank you for your professional comment. In this work, GANT58 was ordered from MedChemExpress (MCE; Cat. No.: HY-13282) Company. According to the instructions for use, we prepared 20 mg/ml ethanol solution of GANT58 into 2 mg/ml working solution for injection in vivo according to the following ratio: 10% EtOH + 90% (20% SBE-β-CD in PBS); Clear solution; Need ultrasonic. During the experiment, GANT58 was injected i.p. at a dose of 20 mg/kg daily for 28 days. With regard to the choice of drug injection concentration, according to the previous literature, most studies used a dose of 50 mg/kg for daily injection [1, 2]. Hereby, we set up concentration gradient intervention (0, 10, 20, and 50 mg/kg) in the preliminary experiment and found that 20 and 50 both had good therapeutic effects. Therefore, according to the consideration of economy and safety, we chose 20 mg/kg as our final intervention concentration.

Reference:

[1] Li G, Deng Y, Li K, Liu Y, Wang L, Wu Z, Chen C, Zhang K, Yu B, Hedgehog Signalling Contributes to Trauma-Induced Tendon Heterotopic Ossification and Regulates Osteogenesis through Antioxidant Pathway in Tendon-Derived Stem Cells, Antioxidants (Basel) 16;11(11) (2022) 2265.

[2] Lauth M, Bergström A, Shimokawa T, Toftgård R, Inhibition of GLI-mediated transcription and tumor cell growth by small-molecule antagonists. Proc Natl Acad Sci U S A. 15;104(20) (2007) 8455-60.

Comment 2. Zoom photos in Figure 1j are not clear. Is GLI1 exclusively expressed in F4/80+ macrophages in synovial tissues?

Many Thanks for your comment. In the revised manuscript, we have improved the resolution of the image for better observation. According to the results, although GLI1 is more expressed in F4/80 positive cells, not all GLI1 proteins are expressed in macrophages, and we can find that some GLI1 positive staining is expressed in other cells. In the follow-up study, we will continue to explore this phenomenon and study the relationship between GLI1 and cells like synovial fibroblasts in RA.

Comment 3. In Figure 2 and 3, the treatment of macrophages with IFNγ+LPS and RANKL enhanced the nuclear translocation of GLI1, suggesting that these stimuli may activate hedgehog signals. Recent studies, however, suggest various non-canonical activation pathways of GLI1. Does hedgehog inhibitor (ex. SMO inhibitor) also suppress M1 polarization and osteoclastogenesis?

Thank you for your constructive comment. We agree with that the activation of GLI1 is regulated by many various pathways. According to your comment, we additionally used Cyclopamine, a selective inhibitor of SMO, to intervene during the polarization of M1 macrophages and the activation of osteoclasts. The results are as follows: Cyclopamine could also inhibit the pro-inflammatory polarization of macrophages to a certain extent, and a significant inhibition of the osteoclast formation could be observed as well. These results may further confirm the important role of HH/GLI1 in regulating macrophage caused inflammation and osteoclast activation.

**Author response image 2. sa2fig2:** (a) Immunofluorescence staining of iNOS in RAW264. 7 cells during the stimulation by LPS/IFN-γ with or without 10 μM cyclopamine intervention and mean fluorescence intensity of the iNOS immunofluorescence. (b) TRAP staining of RAW264.7 cells after induction by RANKL with or without 10 μM cyclopamine intervention and TRAP positive stained osteoclasts quantification. Data shown represent the mean ± SD. Statistical analysis was performed using one-way ANOVA test. *p < 0.05, **p < 0.01, ns = no significance.

Comment 4. In Figure 6, the overexpression of DNMT3a reversed the inhibitory effects of GANT58 in osteoclastogenesis. This supports the author's conclusion that GLI1 may enhance osteoclastogenesis via DNMT3a upregulation. However, this conclusion should be carefully evaluated by examining effects of the overexpression of DNMT3a without GANT58. Does the overexpression of DNMT3a by itself enhance osteoclastogenesis or just reverse the GANT58-mediated suppression?

Thanks for your constructive comment. According to your comment, we observed and compared the differences in the activation of osteoclasts between the DNMT3a overexpression group and the control group. The results showed that at day 4 of induction, the TRAP positive stained osteoclast number seemed to be no significance between WT group and *Dnmt3a^OE^* group. However, at day 3, there was more osteoclast in *Dnmt3a^OE^* group, which suggested that overexpression of *Dnmt3a* might accelerate the activation of osteoclasts to some extent. The results are shown in Figure 6—figure supplement 2.

Comment 5. Is RNA-seq data with GANT58 compatible with known target genes of GLI1 reported in previous studies?

Thanks for your constructive comment. By consulting and comparing with other research articles, most of the data trends in RNA sequencing results are the same as those in other studies. In addition, the expression of some genes is different from other studies (MMP13 increased in our data but decreased in other study [1]), which may be caused by different cell lines and different intervention methods.

Reference:

[1] Akhtar N, Makki MS, Haqqi TM, MicroRNA-602 and microRNA-608 regulate sonic hedgehog expression via target sites in the coding region in human chondrocytes, Arthritis Rheumatol 67(2) (2015) 423-34.

Reviewer #3 (Significance (Required)):SignificanceThe main limitation of this paper is the lack of siRNA knockdown study of GLI1 and DNMTs. Another limitation of this paper is that the direct in vivo data demonstrating the inhibitory effects of GANT58 on M1 macrophage and osteoclast activation in CIA is lacking. The strength is the promising activity of GLI inhibitor, GANT58 as an anti-rheumatic drug on monocyte/macrophage-associated inflammation and bone destruction. The roles of hedgehog/GLI signals in macrophage function are largely unknown, and the findings of this study may contribute to this research field. This study will be interesting to rheumatologists and immunologists.

Thanks again for your constructive comments, which helped us to improve the quality of the manuscript.